# Phase separation of a microtubule plus-end tracking protein into a fluid fractal network

Mateusz P. Czub[1], Federico Uliana [2,3], Tarik Grubić [1], Celestino Padeste [1], Kathryn A. Rosowski[4,11], Charlotta Lorenz[4,5], Eric R. Dufresne [4,6], Andreas Menzel [7], Ioannis Vakonakis[8,12], Urs Gasser [9] & Michel O. Steinmetz [1,10] ✉

Microtubule plus-end tracking proteins (+TIPs) participate in nearly all microtubule-based cellular processes and have recently been proposed to function as liquid condensates. However, their formation and internal organization remain poorly understood. Here, we have study the phase separation of Bik1, a CLIP-170 family member and key +TIP involved in budding yeast cell division. Bik1 is a dimer with a rod-shaped conformation primarily defined by its central coiled-coil domain. Its liquid condensation likely involves the formation of higher-order oligomers that phase separate in a manner dependent on the protein's N-terminal CAP-Gly domain and C-terminal EEY/F-like motif. This process is accompanied by conformational rearrangements in Bik1, leading to at least a two-fold increase in multivalent interactions between its folded and disordered domains. Unlike classical liquids, Bik1 condensates exhibit a heterogeneous, fractal supramolecular structure with protein- and solvent-rich regions. This structural evidence supports recent percolation-based models of biomolecular condensates. Together, our findings offer insights into the structure, dynamic rearrangement, and organization of a complex, oligomeric, and multidomain protein in both dilute and condensed states. Our experimental framework can be applied to other biomolecular condensates, including more complex +TIP networks.

Microtubule plus-end tracking proteins (+TIPs) are a diverse group of multidomain, often oligomeric, proteins that localize to and track microtubule plus ends where they form complex and dynamic protein interaction networks[1]. These +TIP networks are well conserved across species, spanning from yeast to humans. Depending on their capacity to track microtubule ends independently, +TIPs can be categorized into two distinct groups. Autonomous +TIPs, like the members of the end binding (EB) protein family, bind directly to microtubule plus ends and belong to the most conserved and ubiquitous +TIPs[2]. The second class of +TIPs consists of proteins that localize to microtubule ends through direct or indirect interactions with EBs[3]. Prominent members are the cytoplasmic linker protein 170 (CLIP-170), the adenomatous polyposis tumor suppressor (APC), SLAINs, and the microtubule-actin crosslinking factor (MACF)[4,5]. Notably, +TIPs and their interaction networks play crucial roles in regulating microtubule dynamics and, as such, participate in virtually

[1]PSI Center for Life Sciences, Villigen PSI, Switzerland. [2]Institute of Biochemistry, Department of Biology, ETH Zürich, Zürich, Switzerland. [3]Johannes Gutenberg University of Mainz, Mainz, Germany. [4]Department of Materials, ETH Zürich, Zürich, Switzerland. [5]Department of Biochemistry, University of Zurich, Zurich, Switzerland. [6]Department of Materials Science and Engineering, Department of Physics, Cornell University, Ithaca, NY, USA. [7]PSI Center for Photon Science, Villigen PSI, Switzerland. [8]Department of Biochemistry, University of Oxford, Oxford, UK. [9]PSI Center for Neutron and Muon Sciences, Villigen PSI, Switzerland. [10]University of Basel, Biozentrum, Basel, Switzerland. [11]Present address: Roche Pharma Research and Early Development, F. Hoffmann-La Roche Ltd, Basel, Switzerland. [12]Present address: Lonza Biologics, Microbial Development, Visp, Switzerland. ✉e-mail: michel.steinmetz@psi.ch

all microtubule-based cellular processes, including cell division, cell motility, and intracellular signaling[4].

An example illustrating the key implication of +TIPs is well seen in the budding yeast *Saccharomyces cerevisiae*, where the Kar9-mediated +TIP network drives nuclear positioning during mitosis and mating[6]. The Kar9 network consists of the core proteins Kar9 (functional homolog of APC, SLAINs, and MACF), Bim1 (EB orthologue), and Bik1 (CLIP-170 orthologue). Specific protein interactions within the Kar9 network have been characterized in the past in large detail[6–9], and recent studies have revealed that Kar9, Bim1, and Bik1 exhibit liquid-liquid phase separation (reviewed in refs. [10–13]) when combined in vitro[14]. In this context, Bik1 has been shown to be the key driver behind the liquid condensation of this protein trio, significantly reducing the critical concentration needed for droplet formation in the mixture[14]. Importantly, the Kar9 network has been suggested to also form a liquid condensate in vivo, dubbed the "+TIP body", which tracks dynamic microtubule ends and facilitates force transmission between actin cables and microtubule ends during mitosis via the F-actin-directed myosin motor Myo2[14]. The formation and functioning of the Kar9 +TIP body are predominantly influenced by folded domains, linear motifs present in disordered protein regions, oligomerization domains, and great multivalency and redundancy between interaction partners[14]. Notably, similar protein elements were reported to mediate the phase separation of +TIPs in other eukaryotic organisms, including EBs and CLIP-170[15–17]. However, the complexity of +TIP condensates currently hinders our comprehensive understanding of their properties, organization, and modes of regulation.

Bik1 is critically involved in the regulation of microtubule dynamics and participates in nuclear fusion, chromosome disjunction, and nuclear segregation during mitosis in *Saccharomyces cerevisiae*[18]. Interestingly, Bik1 was recently reported to undergo condensation in vitro in reduced ionic strength conditions (change from 500 to 250 mM sodium chloride concentration) without the need of mixing it with other protein partners or crowding agents[14]. Intriguingly, micrometer-sized Bik1 droplets have an unusually low density and high viscosity compared to other condensates, including FUS[19].

Here, to deepen our understanding of the formation process and internal organization of +TIP condensates, we focused on Bik1 due to its intriguing structural and biophysical properties. We employed a combination of computational modeling, microscopy, small-angle X-ray scattering (SAXS), and crosslinking mass spectrometry (XL-MS) to investigate the phase separation of Bik1. This integrative approach provides a comprehensive view of Bik1's structure, dynamics, and interactions, delivering mechanistic insights into the phase separation process of this complex, oligomeric, and multidomain protein.

## Results

### Domain organization, predicted structure, and phase separation of Bik1

To obtain structural information on the Bik1 homodimer[7,20], we predicted its structure using AlphaFold[21]. Based on this computational approach, Bik1 could form an elongated structure with the following domain organization (Fig. 1A): an N-terminal cytoskeleton-associated protein glycine-rich (CAP-Gly) domain (residues 1–80), a flexible linker region (L1, residues 81–188), a central two-stranded, parallel, in-register coiled coil (residues 189–389), and a C-terminal domain (residues 390–440) containing a short flexible linker region (L2, residues 390–419), a zinc-finger domain (residues 420–431), and a flexible tail region (residues 432–440) containing the C-terminal EEY/F-like linear motif QFF (ETF in CLIP-170)[22,23]. The CAP-Gly domain of Bik1, whose structure has been solved by X-ray crystallography[7], mediates the interaction with the C-terminal EEY/F-motif of Bim1 (ETF)[7,24] and likely with the one of α-tubulin (EEF)[25]. These interactions were shown to play a role in Bik1 localization to microtubule plus ends and in regulating microtubule assembly and dynamics[7,26]. It has not yet been investigated whether the N-terminal, CAP-Gly containing domain of Bik1 can bind to its own C-terminal domain as has been found for CLIP-170[27].

While alternative AlphaFold predicted, dimeric structures of Bik1 align well with the proteins' domain boundaries, they significantly diverge in terms of overall molecular shapes (Supplementary Fig. 1). Notably, the coiled-coil domain is predicted by AlphaFold and DeepCoil[28] to consist of three segments (segment A: residues 189–297; segment B: 302–357; segment C: 363–389) that are interrupted by discontinuities in the heptad-repeat sequence (residues 298–301 between segments A and B (A–B linker), and residues 358–362 between segments B and C (B–C linker; Fig. 1A and Supplementary Fig. 1B). These coiled-coil segment linkers act as hinges that allow the protein to adopt either an elongated or "V-shaped", folded-back conformations. The CAP-Gly domain is connected to the coiled coil via a flexible linker and as shown in the various AlphaFold predicted structures, can be located at different distances from the coiled-coil domain. The CAP-Gly domain surface is predominantly positively charged, while other domains and regions of Bik1 contain several positively and negatively charged patches along the elongated structure of the protein (Fig. 1B).

To assess the phase separation of N-terminally His-tagged, full-length Bik1 we recombinantly expressed and purified the protein from bacteria (Bik1 FL; Fig. 1C, Supplementary Fig. 2A). The circular dichroism (CD) spectrum and thermal unfolding profile recorded at 222 nm for Bik1 FL were characteristic of well-folded, α-helical structures (Supplementary Fig. 2B, C)[29]. As shown in Fig. 1D, the phase separation of Bik1 FL depends on both protein and salt concentrations and gives rise to the formation of micrometer-sized droplets at a protein concentration of 5 μM and in the presence of 250 mM sodium chloride concentration (Supplementary Fig. 3), which have been previously observed to merge over time (liquid droplet-like behavior)[14]. Notably, the presence of the N-terminal His-tag did not affect the protein's behavior or its ability to undergo phase separation under the conditions tested (Supplementary Fig. 3). We therefore retained the His-tag on all constructs in subsequent experiments.

Confocal fluorescence microscopy experiments with an N-terminal His-mNeonGreen fusion version of Bik1 FL (mNG-Bik1 FL) were performed to investigate the protein distribution within Bik1 droplets. These fluorescence microscopy experiments suggest that the distribution of Bik1 molecules is uniform throughout the droplet; the fluorescence of Bik1 is over 60 times brighter in the condensed than in the dilute phase under the condition tried (Fig. 1E, F). From these experiments, we can conclude that Bik1 readily phase separates under low salt conditions and in the low micromolar protein concentration range. The overall intracellular Bik1 concentration in a budding yeast cell can be estimated to be in the 100 nanomolar range[30]; however, we expect that its local concentration is significantly higher when Bik1 localizes at the tip of a microtubule. Our in vitro phase separation experiments may thus roughly approximate the protein's concentration range present at microtubule tips in vivo.

Next, to assess which parts of Bik1 are required for the protein to undergo phase separation, we prepared three Bik1 truncation mutants (Fig. 1C): Bik1 ΔN, lacking the N-terminal CAP-Gly domain and the linker L1; Bik1 ΔNC, lacking both the N- and C-terminal regions and essentially corresponding to Bik1's coiled-coil domain; and Bik1 ΔQQFF, lacking the C-terminal EEY/F-like motif that is crucial for CLIP-170 to interact with the CAP-Gly domain of p150glued[31]. As shown in Supplementary Fig. 3, none of the three Bik1 fragments formed droplets even at a ten-fold higher protein concentration (50 μM) and five-fold lower sodium chloride concentration (50 mM) as needed for Bik1 FL condensation. We also investigated whether mixing Bik1 ΔN and Bik1 ΔQQFF can rescue their potential to undergo condensation at very high protein and very low salt concentrations; however, no droplets were observed either (Supplementary Fig. 3).

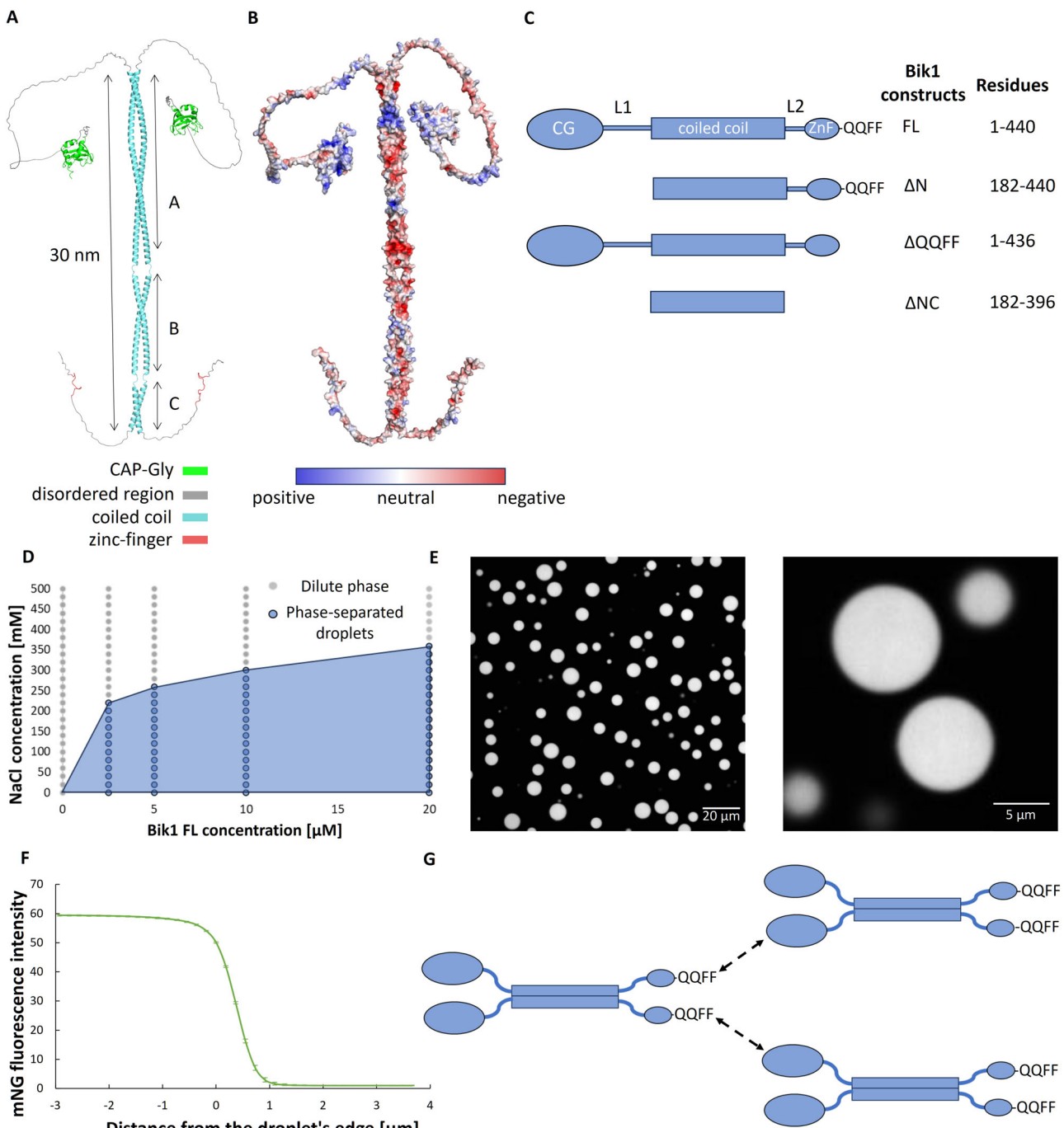

**Fig. 1 | Structural organization and phase separation of Bik1. A** One possible structure of the Bik1 dimer as predicted by AlphaFold (for alternative predicted structures, see Supplementary Fig. 1A) with color-coded domains. The three predicted coiled-coil segments A, B, and C of the coiled-coil domain, which are separated by linkers are indicated. **B** Charge distribution on Bik1's surface. Positively charged patches are colored in blue, negatively charged patches are in red, and neutral patches are in light gray. **C** Bik1 truncation mutants used in this study with their respective designations and residue boundaries. CG, CAP-Gly domain; L1 and L2, linker regions 1 and 2; FL, full-length Bik1; ΔN, Bik1 variant lacking the N-terminal CAP-Gly domain and the linker L1; ZnF, zinc finger; QQFF, peptide Gln-Gln-Phe-Phe; ΔNC, Bik1 variant lacking both the N- and C-terminal regions and corresponding to Bik1's coiled-coil domain. **D** Phase diagram of N-terminally His-tagged Bik1 FL in

20 mM Tris-HCl, pH 7.4, supplemented with 1 mM DTT and 10% glycerol. **E** Confocal fluorescence microscopy images of micrometer-sized droplets observed for 40 µM Bik1 (mixture of 90% Bik1 FL and 10% mNG-Bik1 FL; both proteins are N-terminally His-tagged) in a buffer consisting of 20 mM Tris-HCl, pH 7.4, supplemented with 250 mM NaCl and 2% glycerol. View from the top (left) and a mid-plane slice (right). **F** The average intensity ($N = 233$) of mNG throughout mNG-Bik1 FL droplets is shown in (**E**). Error bars represent standard errors. **G** Possible inter-molecular interactions between Bik1 dimers mediated by the protein's N-terminal CAP-Gly domain and C-terminal EEY/F-like motif. Unstructured regions of the protein are shown as blue solid lines. CAP-Gly, coiled coil, and zinc-finger domains are depicted in the same way as in (**C**). Source data of **D** and **F** are provided in the Source Data file.

These findings collectively suggest that Bik1 is a complex, multi-domain homodimer consisting of several folded domains and sub-domains that are separated by flexible linker regions. Additionally, they indicate that a possible interaction between the CAP-Gly domain and the EEY/F-like motif of Bik1 is key for phase separation into liquid droplets (Fig. 1G).

## Size-exclusion chromatography followed by small-angle X-ray scattering analysis of Bik1

To experimentally assess the conformation of the Bik1 FL dimer in solution, we performed size-exclusion chromatography followed by small-angle X-ray scattering (SEC-SAXS) experiments[32]. In addition, we analyzed Bik1 ΔQQFF, which is not able to undergo phase separation, and the coiled-coil domain of Bik1 (Bik1 ΔNC; Fig. 1C) to ascertain its contribution to the overall conformation of Bik1 FL and Bik1 ΔQQFF.

Examples of scattering profiles obtained in the high-salt buffer (500 mM sodium chloride) recorded for all three variants are shown in Fig. 2A. The pair distance distribution function, P(r), revealed similar values of radius of gyration, Rg, of 88–98 Å for Bik1 FL, 94–97 Å for Bik1 ΔQQFF, and 79–83 Å for Bik1 ΔNC, as well as similar values of the maximal particle dimension, Dmax, for all variants of 280–320 Å, 294–320 Å, and 250–298 Å, respectively (Fig. 2B, Supplementary Tables 1 and 2). Notably, the Dmax values are consistent with the predicted length of the fully elongated Bik1 coiled-coil domain (~300 Å; Fig. 1A). Kratky analysis of the data (Supplementary Fig. 4A) suggests that the measured proteins are elongated and may include some unstructured elements.

The bead-based envelope of the Bik1 FL dimer at a resolution of $61 \pm 4$ Å was calculated ab initio by averaging the 19 most similar models (out of 20 generated), resulting in a normalized spatial discrepancy (NSD) value of $0.72 \pm 0.06$, indicating a level of convergence in the model calculations (Fig. 2C). Similar envelopes were calculated for Bik1 ΔQQFF (resolution $65 \pm 5$ Å; NSD value of $0.71 \pm 0.05$) and Bik1 ΔNC (resolution $43 \pm 3$; NSD value of $0.61 \pm 0.04$) by averaging 19 models (out of 20 generated) for each variant (Supplementary Fig. 4B, C). The molecular envelope of Bik1 FL indicates that the Bik1 dimer adopts an elongated conformation with presumably two distinctive kinks, one in the middle part of the envelope and another one at its extremity. These two kinks could potentially be attributed to the predicted flexible A-B and B-C linkers between coiled-coil segments (Fig. 1A, Supplementary Fig. 1). Notably, the envelope obtained for Bik1 ΔNC also displays two kinks at similar positions (Fig. 2D, Supplementary Fig. 4), implying that the coiled-coil domain of Bik1 primarily defines the overall elongated shape of Bik1 FL's envelope. The connection between the CAP-Gly domain and the coiled coil is mediated by the unstructured linker L1 that holds the potential for diverse spatial arrangements. Consequently, akin to the C-terminal domain, these regions do not significantly contribute to the averaged overall shape of the calculated envelope.

Together, these findings are consistent with full-length Bik1 being a predominantly elongated dimer in solution, whose overall conformation is defined by its two-stranded, parallel coiled-coil domain. They further indicate that both the N-terminal CAP-Gly and C-terminal zinc-finger and EEY/F-like motif-containing domains are connected to the coiled coil via the flexible linkers L1 and L2.

## Qualitative crosslinking mass spectrometry of Bik1 in its dilute and condensed states

To investigate differences in interactions, conformations, and dynamics of Bik1 FL between its dilute and condensed states, we employed a qualitative chemical crosslinking followed by a mass spectrometry (XL-MS) approach[33,34]. To this end, we used two crosslinking reagents, disuccinimidyl suberate (DSS) and pimelic acid dihydrazide (PDH), which react with primary amines and acidic amino acids, respectively[35,36]. During the experiments, we identified the following four types of crosslinked peptides: (i) "crosslinks" between two different residues; (ii) "monolinks" between a single amino acid and the crosslinking reagent; (iii) "selflinks" between two positionally identical amino acids; (iv) "zerolinks" between amino acids that are covalently bonded without linker (only conditions with PDH).

Using DSS, we performed crosslinking experiments (three independent replicates) followed by trypsin digestions with "dilute" single-phase (500 mM sodium chloride) and "phase separated" (250 mM sodium chloride) Bik1 FL. We further performed similar experiments with the "pellet" and "supernatant" fractions obtained after centrifugation of phase-separated samples. Since Bik1 forms homodimers in solution, we were not able to distinguish between intra- and intermolecular crosslinks, except for selflinks that are always intermolecular and thus can be used to describe the protein's oligomerization state. Employing a conservative filtering strategy based on spectra quality (ld.Score >20) and consistency (crosslinked peptides must be identified in at least two out of the three independent replicates), we identified 79 unique crosslinked peptides for phase-separated Bik1 FL (37 crosslinks, 36 monolinks, and 6 selflinks) and 65 unique crosslinked peptides for dilute Bik1 FL (33 crosslinks, 30 monolinks, and 2 selflinks; Fig. 3A, Supplementary Data File 1). To test for potential artifacts, different crosslinking reagent concentrations, and distinct fragmentation conditions were tested. As shown in Supplementary Fig. 5AB and Supplementary Data File 2, we obtained similar results independent of the experimental conditions used, highlighting the robustness of the method.

As shown in Fig. 3B, approximately half of the obtained crosslinked peptides are shared across all conditions tested, with the pellet condition exhibiting 14 unique ones, while the pairs phase separated/pellet and dilute/supernatant shared 9 and 4 crosslinked peptides, respectively. Differential analysis of spectra indicates an increased number of crosslinks and selflinks in phase-separated Bik1 FL samples, while the opposite trend was found for monolinks (Fig. 3C). Remarkably, the same pattern can also be seen in the supernatant and pellet conditions (Fig. 3C). The observation of an increased number of self-link peptides combined with the reduced number of monolink peptides that are hydrolyzed by water molecules indicate that during the transition from dilute to phase separation, Bik1 FL adapts a more compact conformation (crosslinked residues are located closer in sequence) with an increased tendency to form higher-order oligomers.

The crosslinks identified using DSS chemistry cover most of the Bik1 sequence, excluding its C-terminal region encompassing the zinc-finger domain and the QQFF motif. This region is marked by high flexibility, numerous acidic residues, and a lack of lysine and arginine residues that are targeted by DSS. To address this limitation, we switched to the PDH crosslinking reagent in combination with different proteases (AspN, GluC, chymotrypsin, and a combination of AspN and GluC). We screened different conditions in single experiments and using a threshold of ld.Score >25 and FDR < 0.05, we identified 14 unique crosslinked peptides in the phase-separated condition and 44 for the dilute phase; several of these peptides connect the protein's coiled coil and C-terminal regions (Fig. 3D, Supplementary Data File 1).

Next, we assessed the number of spectra assigned to PDH and DSS crosslinked peptides in the context of the four computationally identified Bik1 domains and regions (i.e., CAP-Gly, disordered region L1, coiled coil, and C-terminal region containing disordered region L2 and the zinc finger; Fig. 1A) to pinpoint which of those are more interconnected and which ones undergo rearrangements during the protein's transitioning from the dilute to the condensed state. We performed this analysis by summing up all identified spectra (Fig. 3D, E) or by calculating the average number of spectra per condition considering only DSS crosslinked peptides (Supplementary Fig. 6A, B), which led to the following observations.

Firstly, when comparing the phase separation and dilute conditions, we observed an increase in selflinks within the coiled-coil

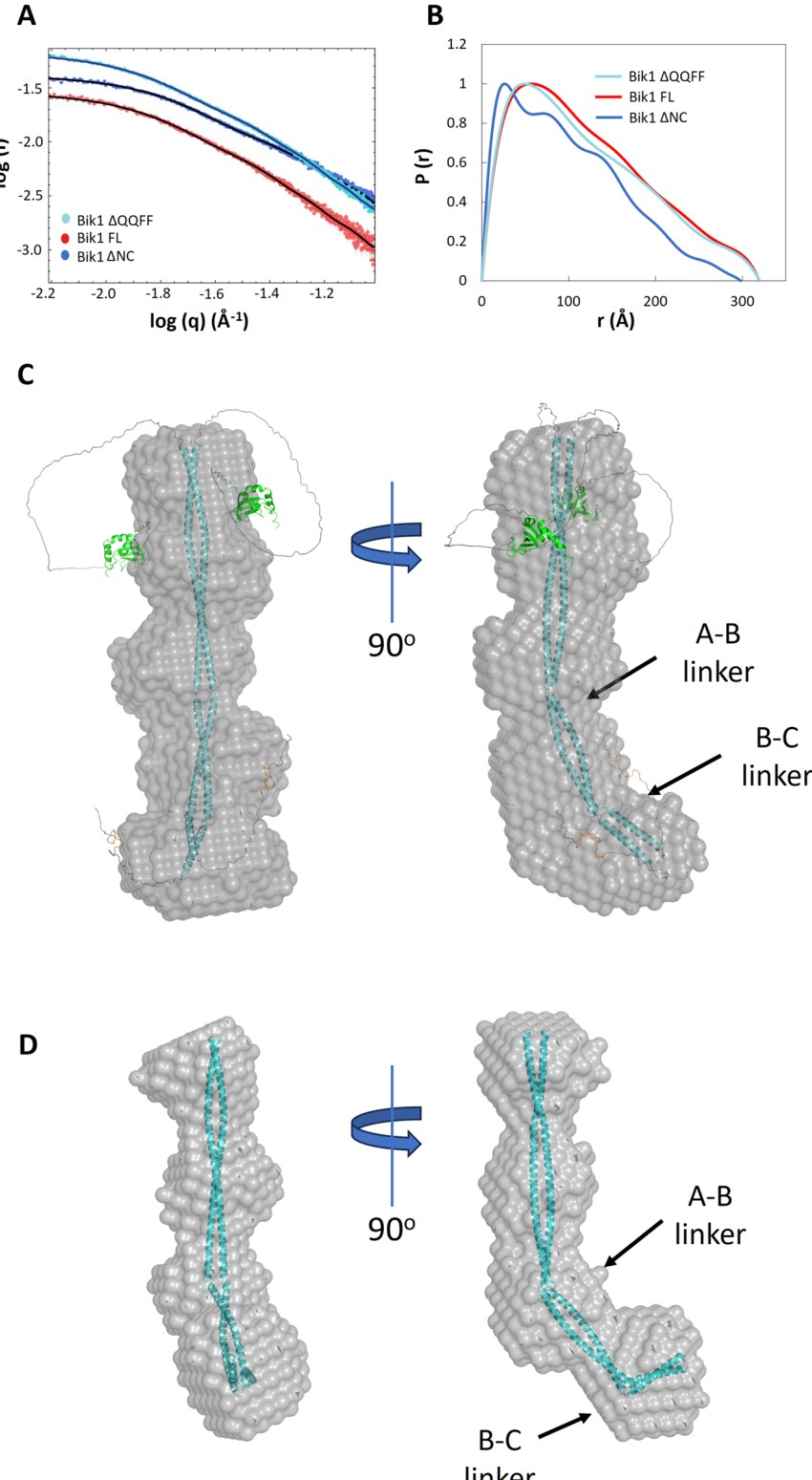

**Fig. 2 | SEC-SAXS analysis of Bik1 variants. A** Representative scattering profiles obtained for Bik1 FL, Bik1 ΔQQFF, and Bik1 ΔNC in a high-salt buffer (20 mM Tris-HCl, pH 7.5, supplemented with 500 mM NaCl, 2% glycerol, and 1 mM DTT). Solid black lines represent fits of the generated Bik1 models (shown in **C** and **D** and Supplementary Fig. 4) to the experimental data. **B** Representative distance distribution functions obtained for Bik1 FL, Bik1 ΔQQFF, and Bik1 ΔNC. **C** A DAMMIF-generated molecular envelope (beads model) of Bik1 FL superimposed with its AlphaFold predicted model shown in Fig. 1A and adjusted to fit the beads model. **D** The molecular envelope of Bik1 ΔNC superimposed with the Bik1 AlphaFold model truncated to the coiled coil and adjusted to fit the beads model. Two kinks suggested by the shape of the calculated envelope are marked with black arrows. See also Supplementary Fig. 4.

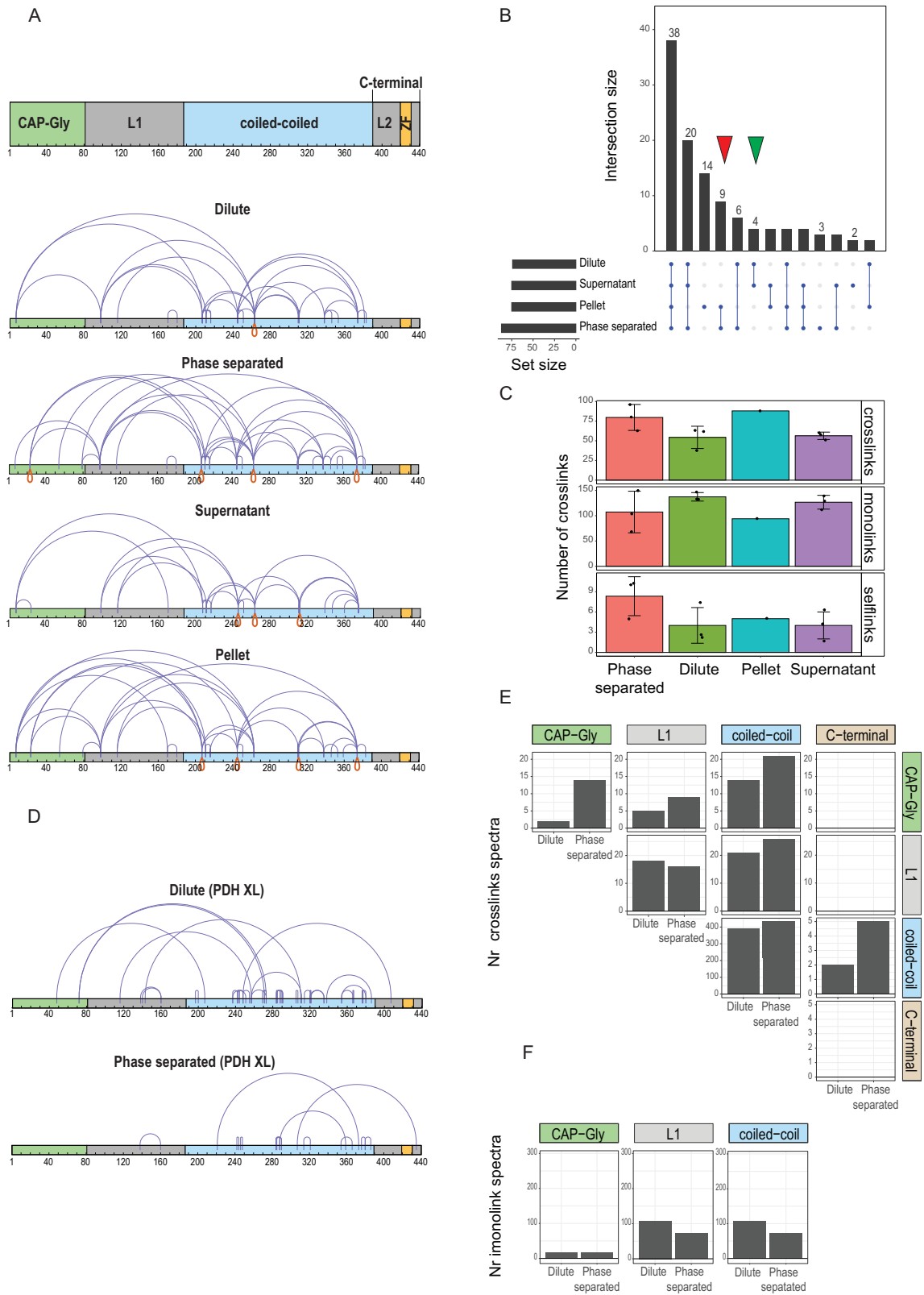

domain (22 versus 12, with peptides 207 and 374 specific for the phase separation condition, Fig. 3A). This finding is further supported by the increased number of crosslinks found in the coiled coil during phase separation (216 versus 195, Fig. 3E). Secondly, while the number of crosslinked peptides stemming from the L1 region is similar for both the phase separated and dilute conditions (9 versus 8), we observed an increase in crosslinked peptides in former condition for those linking

the CAP-Gly domain to either L1 (9 versus 5) or the coiled- oil (21 versus 14), as well as for those linking the C-terminal region with the coiled coil (5 versus 2; Fig. 3E). Thirdly, concomitant with a change in the peptide crosslink pattern, we found a reduced number of monolink peptides stemming from the coiled coil and the L1 region in phase separated compared to dilute conditions (108 versus 72 and 293 versus 233, respectively; Fig. 3F).

**Fig. 3 | Qualitative XL-MS of Bik1 FL. A** DSS crosslinks and selflinks (depicted in blue and in red, respectively) identified in dilute (10 mM HEPES, pH 7.5, supplemented with 500 mM NaCl) and phase-separated (5 mM HEPES, pH 7.5, supplemented with 250 mM NaCl) conditions, and after centrifugation of phase-separated condition (fractionations: supernatant and pellet). Crosslinked peptides were filtered based on the quality of the spectra (Id.Score >20 and consistency, requiring identification in at least two out of three independent experimental replicates). Bik1 domains and regions are shown in the top panel: CAP-Gly domain, green; linker regions L1 and L2, light gray; coiled coil, cyan; zinc-finger domain ZF, orange. **B** Upset plot for filtered DSS crosslinked peptides showing the overlap among the following conditions: dilute, phase separated, supernatant, and pellet. The red and green triangle marks the number of crosslinked peptides shared between the pellet and phase separation and between the dilute and supernatant conditions. **C** Quantification of filtered DSS crosslinked peptides (monolinks, crosslinks, and selflinks) for the following conditions: dilute, phase separated, supernatant, and

pellet. Data are presented as mean values ± SD and each dot represents the results of an independent experiment ($N = 3$, except for the pellet condition which has no replicates) the bar plot shows the average number across three conditions with the error bars representing the standard deviations of three independent experimental replicates. **D** PDH crosslinks identified in dilute, phase-separated conditions. Crosslinked peptides were filtered based on the quality of the spectra (Id.Score >25). The plot shows the PDH crosslinked peptides obtained from different protease conditions (AspN, GluC, AspN+GluC, and chymotrypsin). All conditions have been tested without replicates. **E** Number of identified spectra for crosslinked DSS and PDH peptides associated with Bik1 domains and regions. The bar plot reports the sum of all identified spectra for the dilute and phase-separated conditions. **F** Number of identified spectra for PDH and DSS monolink peptides associated with the Bik1 domain and regions. The bar plot reports the sum of all identified spectra for the dilute and phase-separated conditions. Source data for all panels are provided in the Source Data file.

Taken together, these qualitative XL-MS results support the predicted parallel, in-register organization of the two coiled-coil chains of Bik1 (Fig. 1A, Supplementary Fig. 1A). They also indicate that Bik1 undergoes conformational rearrangements during condensation, involving mainly interactions within the coiled-coil domain or between the coiled coil and its N- and C-terminal flanking regions. These multiple interactions might at least partially be explained by weak, non-specific electrostatic interactions between positively and negatively charged surface patches distributed along the Bik1 dimer (Fig. 1B). Lastly, as the abundance of monolinks correlates with the flexibility and solvent exposure of residues[37], our findings indicate an increased propensity of Bik1 to form a multivalent interaction network in the condensed phase.

## Quantitative crosslinking mass spectrometry of Bik1 variants

To get a deeper insight into the findings obtained by qualitative XL-MS, we quantitatively measured the intensity of the identified crosslinked Bik1 FL peptides[38–40]. As Bik1 molecules can be in a constant exchange between the dilute and condensed phase, and as the crosslinking reaction is not selective for one distinct state, the approach used will characterize a heterogeneous system[41]. To overcome this limitation, we followed a reductionist, two-step approach: (i) identification of peptides that are specific for one protein state (referred to as "conformo-specific" peptides); (ii) quantification of crosslinked peptides exploiting sensitivity and high accuracy of targeted proteomics[42,43] (Fig. 4A). For instance, we classified the crosslinked peptides 211–346, 211–359, 216–245, 79–338, and 245–346 as conformo-specific crosslinks, as they were identified only in the pellet condition. We complemented the list with four self-linked peptides identified in the coiled-coil domain of Bik1 to probe interactions promoting subunit dimerization (207–207, 311–311, 338–338, and 374–374; Fig. 4D, Supplementary Data File 3, Supplementary Table 3). To quantify crosslinked peptides, we repeated the experiment by crosslinking samples with DSS followed by trypsin digestion. The intensity of targeted peptides was then normalized for the abundance of protein to account for potential level variations by following a Bik1 coiled-coil peptide (residues 224-235; Supplementary Fig. 7A). All experiments were conducted in three independent replicates (Supplementary Data File 3, Supplementary Fig. 7B, D).

In the first experiment, we compared the pellet and supernatant conditions of phase-separated Bik1 FL (150 mM sodium chloride). As shown in Fig. 4B, the differential analysis confirmed that all crosslinks identified exclusively in the pellet of the qualitative XL-MS experiments were enriched at least by a factor of two (211–346, 211–359, 216–245, 79–338, 245–346). Furthermore, a differential abundance of selflinks (207–207, 311–311, 338–338, 374–374) shows that there are stronger interactions within the coiled-coil domain in the pellet condition. We then sought to verify whether the crosslinked peptides that are more abundant in the pellet were also significantly enriched when

comparing the phase separation and dilute Bik1 FL conditions (500 mM sodium chloride; Fig. 4C). Although the changes are less evident, these results indicate that the interactions driving Bik1 FL condensation are conserved: they include crosslinked peptides within the coiled-coil domain (207–207, 211–359, 211–346, 216–245) and peptides between the coiled coil and the CAP-Gly domain (79–338, 23–374; Supplementary Fig. 7C). Unlike the pellet condition, where all measured self-linked peptides in the coiled-coil domain are enriched, in phase-separated Bik1 only the 207–207 selflink is enriched.

Next, we analyzed the targeted crosslinked peptides in Bik1 FL, Bik1 ΔN, Bik1 ΔNC, and Bik1 ΔQQFF at high and low sodium chloride concentrations (Fig. 4D, Supplementary Fig. 7D). Remarkably, in all variants except Bik1 ΔQQFF the crosslinked peptides enriched in the phase-separated Bik1 FL condition (log2FC > 1; 207–207, 211–359, 79–338, 211–347, 216–245, 23–374) are either unaffected or much less influenced by the decrease in salt concentration (Fig. 4D). This suggests that these interactions drive phase separation only in the context of the full-length protein. However interestingly, Bik1 ΔQQFF which does not form droplets even at very high protein and very low salt concentrations (Supplementary Fig. 3) exhibits an intermediate behavior compared to Bik1 FL. While this minimally truncated Bik1 variant exhibits no significant or small abundance changes for peptides linking the C-terminal part of the coil–coil domain (211–359, 79–338, 211–346, 23–374), the selflink interactions within the coiled coil (207–207 and 374–374) appear to be affected by lowering the salt concentration similarly to what is observed for Bik1 FL (Fig. 4D).

Collectively, our qualitative and quantitative XL-MS results provide insights into the conformational rearrangements of the Bik1 dimer as it transitions from the dilute to the phase-separated state. The findings underscore the critical roles of various regions within Bik1's coiled-coil domain and interactions with its N- and C-terminal flanking regions. This allows the protein to establish a complex, multivalent network of interactions that seem critical for higher-order oligomer formation. Coupled with our microscopy-based phase separation experiments, the data also suggest that the C-terminal EEY/F-like motif is important for the further condensation of these Bik1 oligomers into micrometer-sized liquid droplets.

## Small-angle X-ray scattering analysis of phase-separated Bik1

Next, we investigated the supramolecular structure of Bik1 condensates. To this end, SAXS measurements were conducted on phase-separated Bik1 FL droplets as they flowed through a microfluidic chip (Supplementary Fig. 8A-C). Bik1 droplets showed enhanced scattering at low $q$ (large length scales), which was absent from the dilute SEC-SAXS data (Fig. 5A, Supplementary Fig. 8D). In this regime, the SAXS signal decays as $I(q) \propto q^{-2}$, suggesting the presence of a fractal network with a fractal dimension $d \approx 2$. This fractal structure extends from the lowest $q$ reached in the SAXS data (from >3000 Å) down to about

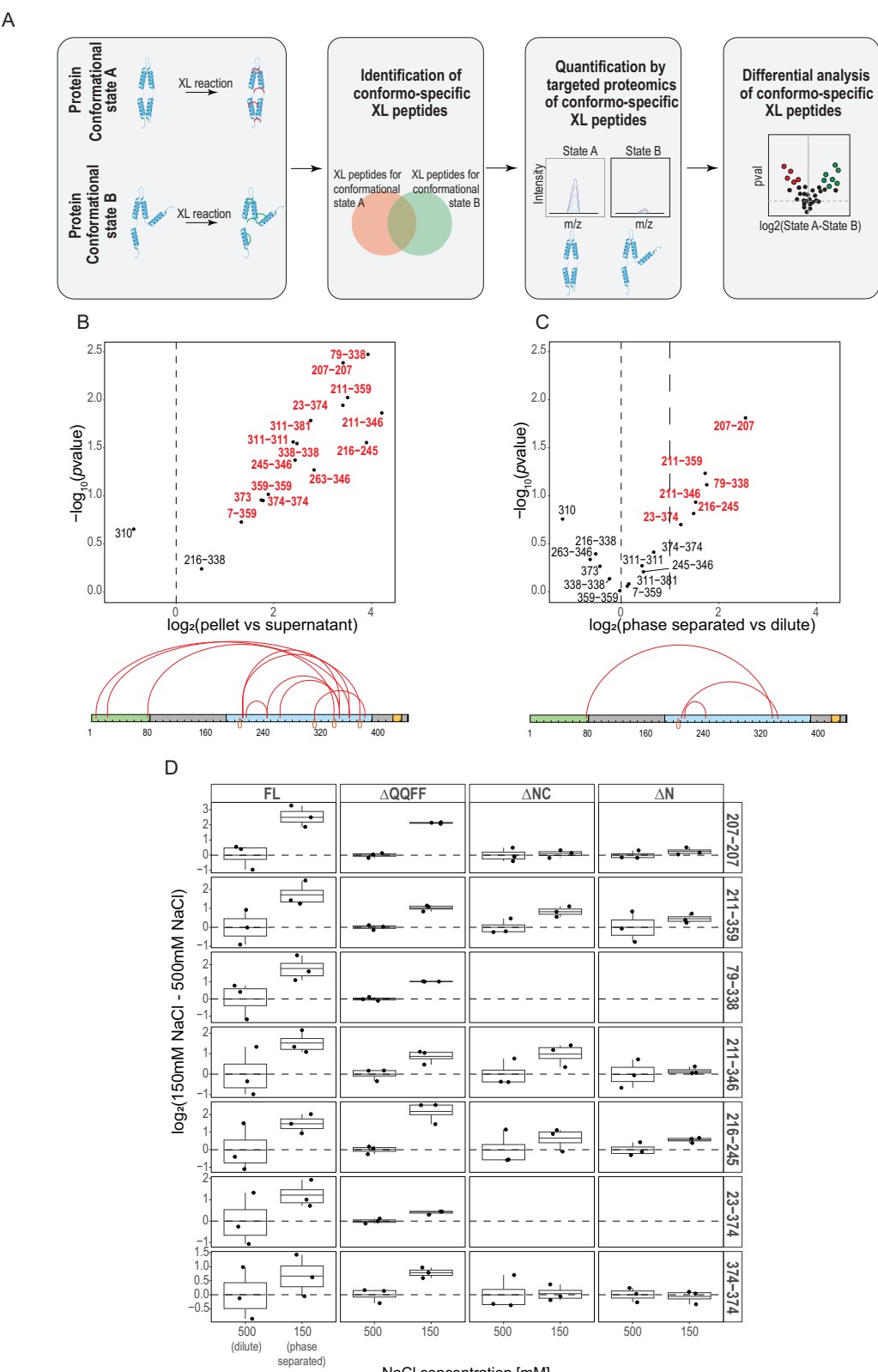

300 Å (Fig. 5A, Supplementary Fig. 8D). To visualize this supramolecular structure, we generated multiple real-space structures that were consistent with the scattering data. In all cases, thousands of copies of a simplified Bik1 structural model were placed in a spherical volume with a diameter of 1 μm (see "Methods" section for details). The positions and orientations of these model copies were iteratively repositioned to fit the SAXS data in the range $0.002\,Å^{-1} < q < 0.03\,Å^{-1}$

corresponding to lengths starting from about 200 Å up to 3100 Å (Fig. 5A). We considered a range of Bik1 models, including a Gaussian blob and three different spherocylinder models that approximate the SEC-SAXS solution structure of the protein (Fig. 5B, Supplementary Fig. 8E–I). In all cases, we achieved a good agreement with the scattering data (Fig. 5A, Supplementary Fig. 8D). As expected for a fractal network, we observed a heterogeneous distribution with protein-rich

**Fig. 4 | Quantitative XL-MS of Bik1 variants. A** Quantitative workflow to probe Bik1 conformational changes. Crosslinked (XL) peptides were obtained from Bik1 FL in dilute, phase separation, supernatant, and pellet conditions (Fig. 3A). Crosslinks specific to one condition (conformo-specific crosslinks) are then employed to monitor structural rearrangements in Bik1 samples using quantitative targeted proteomics. **B, C** Differential abundance levels for conformo-specific crosslinks comparing pellet versus supernatant (**B**) and phase separation versus dilute (**C**) conditions. The volcano plot shows the log2 fold changes of crosslinks abundance and their corresponding statistical significance (two-sided unpaired Student's *t*-test, $N = 3$ independent experiments) between the two conditions. Conformo-specific crosslinked peptides with a $log_2FC > 1$ are annotated in red. For domain designations of the Bik1 bar representation, see the top panel in Fig. 3A. **D** Boxplot showing the abundance of selected crosslinked peptides (Supplementary Table 3) normalized for Bik1 abundance in high or low salt conditions. Each dot represents the result of an independent experiment ($N = 3$). The boxplot boundaries indicate the first (Q1, 25%) and third (Q3, 75%) quartiles, while the lower and upper whiskers are defined by $Q1 - 1.5$ IQR and $Q3 + 1.5$ IQR, respectively. The average value is represented by a line across the center of the box. Data are normalized for the average intensities obtained at high salt concentrations. Source data for (**B**–**D**) are provided in the Source Data file.

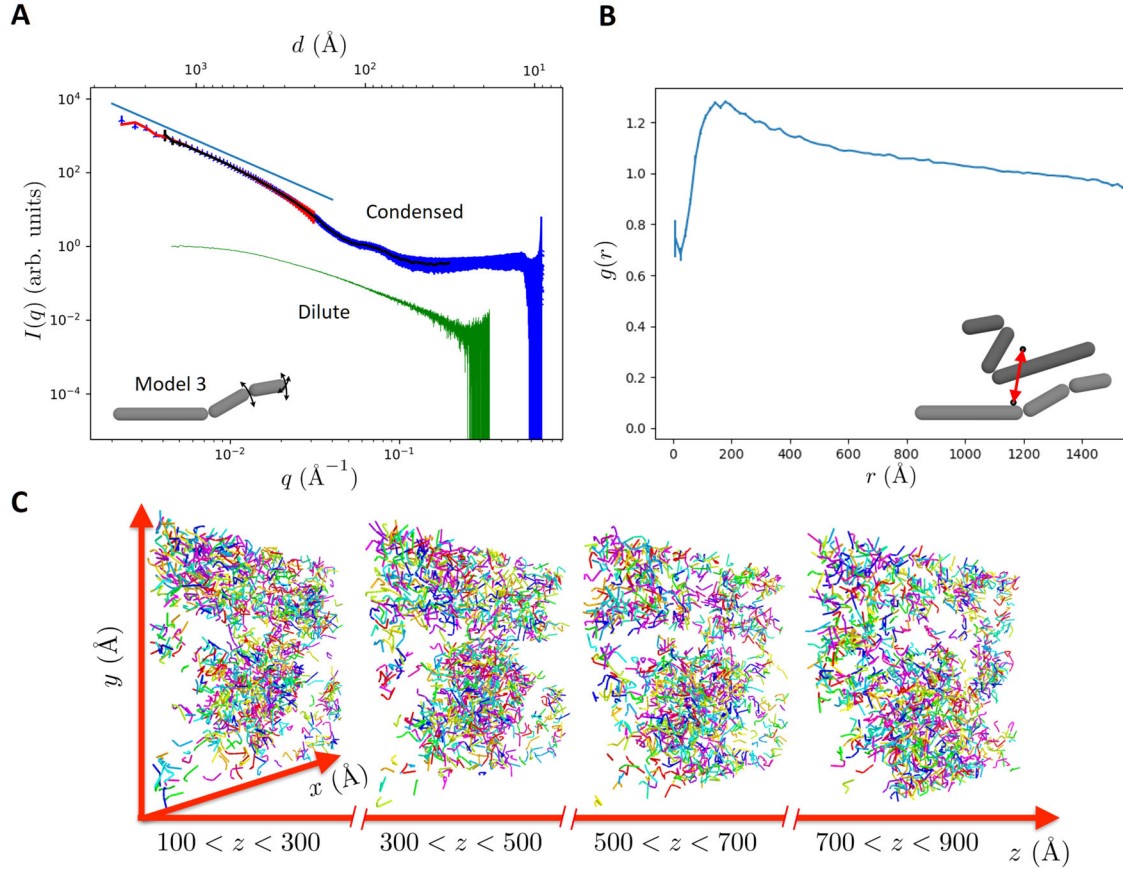

**Fig. 5 | SAXS of phase-separated Bik1. A** SAXS measurement (blue curve) with Gaussian-blob fit (red curve) and fit obtained with 50,000 model 3 copies (black curve). The model 3 fit was used to model the fractal structure and is restricted to $q > 2\pi/\sigma_w$ due to the Gaussian window with $\sigma_w = 1500$ Å. The straight blue line above the SAXS measurement represents the $I(q)$-$q^{-2}$ behavior expected for a fractal network with a fractal dimension of 2. The green curve shows the Bik1 form factor obtained from our SEC-SAXS data (Fig. 2A). Inset, schematic representation of model 3. **B** Radial distribution function of the centers of 50,000 model 3 copies. The decay for $r$ larger than the nearest-neighbor peak is expected for a fractal network. Inset, two model 3 copies with their centers of mass (dots). The red double arrow highlights the distance r between their centers. **C** Sections taken from the fractal structure obtained with 50,000 model 3 copies. Source data for all panels are provided in the Source Data file.

regions separated by solvent-rich voids (Fig. 5C, Supplementary Fig. 8F, Supplementary Movie 1).

To quantify translational order, we calculated the radial distribution function, g(r), from the centers of each Bik1 model in the fitted structure (Fig. 5B). The peak near 200 Å reflects the typical distance between nearest neighbors and the decay at larger distances continues without a transition to a plateau $g(r) = 1$, which is expected for a fractal network. To quantify orientational order, we calculated the nematic order parameter of the fitted supramolecular structure. The nematic order parameter is a local measure of the relative orientation - it is zero in the absence of orientational order when all molecules have random orientations and one when all molecules are aligned in the same direction. We calculated the nematic order parameter in cubic boxes with side lengths of 400 Å. Since the

number of particles in the box fluctuates due to their heterogeneous distribution, we used a lower cutoff of 30 particles in the box to obtain a reliable order parameter. In these boxes, we find the order parameter to be below 0.4 in all cases (Supplementary Fig. 8J, K). This suggests that Bik1 molecules tend to align but nevertheless remain largely in an isotropic state. With an aspect ratio of 12.5, Bik1 molecules could organize into a strongly aligned (liquid crystalline) state at volume fractions above 0.18[44]. This is about a factor of two above the experimentally estimated volume fraction of Bik1 in condensates[19].

Together, these results indicate that phase-separated Bik1 dimers form a fractal network, formed by regions enriched or depleted of protein. They further indicate an isotropic distribution of Bik1 dimers in the condensed phase.

## Discussion

The ubiquitous nature of protein liquid-liquid phase separation has come to the forefront in several studies, elucidating its fundamental involvement in critical cellular processes[14–17,45]. Compelling evidence is now surfacing, shedding light on the integral role of phase separation in shaping human health and its involvement in the onset of diseases[12,46]. However, the molecular mechanisms of liquid-liquid phase separation and structures of condensed proteins remain poorly understood, particularly when it comes to complex systems. Here, we structurally characterized the multidomain and oligomeric +TIP Bik1 in its dilute single-phase and phase-separated states using a combined computational and experimental approach. To the best of our knowledge, this is the only study to correlate biophysical properties with conformational rearrangements and molecular organization, offering a detailed molecular perspective on protein condensation.

To better understand which domains of Bik1 are essential for the protein to undergo condensation, we produced a few truncation variants. Our experiments reveal that both the N-terminal CAP-Gly domain-containing region and the C-terminal EEY/F-like motif are critical for the condensation of Bik1 into liquid droplets. CAP-Gly domains were previously shown to mediate interaction with C-terminal EEY/F motifs found in α-tubulin, EBs, and CLIP-170 proteins[24]. Our results obtained with Bik1 indicate that the interaction between its CAP-Gly domain and EEY/F-like motif might drive the protein's phase separation.

A combination of computational and SAXS methods allowed us to obtain structural information regarding the overall conformation of the Bik1 homodimer in solution. Our results suggest that Bik1 adopts a predominantly elongated conformation with two kinks caused by discontinuities in the heptad-repeat sequence of its two-stranded, parallel coiled-coil domain. The SEC-SAXS-derived beads models obtained for Bik1 FL and Bik1 ΔNC are very similar to each other and, therefore, suggest that the overall shape of the Bik1 dimer is dominated by the elongated conformation of the coiled-coil domain. The N-terminal CAP-Gly and C-terminal zinc-finger and EEY/F-like motif-containing domains were predicted to be connected to the coiled coil via the flexible linkers L1 and L2. They may have the flexibility to interact with different coiled-coil segments, especially patches with opposite charges, or with themselves.

Using qualitative XL-MS, we confirmed multiple interactions of all three coiled-coil segments of Bik1 with both the N- and C-terminal regions of dilute and phase-separated Bik1. Since both regions engage with comparable segments of the coiled coil, they may also have the potential to interact with each other. Our results further revealed differences in crosslinks observed in low and high salt conditions indicating spatial rearrangements of Bik1 FL during condensation. It is well documented that Bik1's mammalian orthologue, CLIP-170, can fold back upon itself due to interactions between its CAP-Gly containing N- and EEY/F-motif-containing C-terminal domains[27,47]. CLIP-170 was also recently reported to form liquid condensates in vitro and in cells[47]. It was hypothesized that phase separation of CLIP-170 facilitates the concentration of several protein partners, including tubulin, at the ends of microtubules, an activity that contributes to the regulation of microtubule dynamics in cells[47]. We anticipate that in budding yeast, Bik1 may exert similar functions to drive cell division[14].

To study the spatial Bik1 FL conformational rearrangements further, we designed and executed a quantitative XL-MS approach measuring crosslinked peptides identified in only one specific condition (conformo-specific peptides). This approach revealed that interactions between coiled-coil segments or between the CAP-Gly domain and the coiled coil are at least two times more abundant in the condensed compared to the dilute state of the protein. The conformo-specific crosslink intensities in the Bik1 ΔN and Bik1 ΔNC variants remain unchanged between high and low salt concentrations. However, and interestingly, Bik1 ΔQQFF exhibits consistent changes although these were less pronounced compared to Bik1 FL. Since Bik1 ΔQQFF does not

phase separate even at very high protein and very low salt concentrations, these findings collectively suggest that while this minimally truncated variant can form higher-order oligomers in solution, it lacks the ability to undergo the further condensation required for the formation of micrometer-sized, phase-separated droplets.

Finally, we investigated the internal structure of phase-separated Bik1 droplets. Using confocal fluorescence microscopy, we observed that the fluorescently labeled Bik1 molecules are homogenously dispersed at the micron-scale. However, our SAXS data suggests that the Bik1 molecule within droplets is heterogeneously organized at the supramolecular scale, from 30 to 300 nm. At this level of organization, Bik1 condensates form a network with a fractal dimension of about 2, a robust result that we obtained independently of the models used to fit the data (i.e., Gaussian blobs or differently segmented spherocylinder models). We did fit the SAXS data up to $q = 0.2 \, \text{Å}^{-1}$ and the model allowed us to obtain good agreement with the measurements. At length scales below 50 Å, the reliability of the fit decreased and the fit does not allow for detailed conclusions about the structure. This is due to the behavior of the model form factor based on spherocylinders, which drops off more sharply than the measured form factor at $q > 0.15 \, \text{Å}^{-1}$. Thus, the model does not capture all structural features of Bik1 at these small length scales.

Fractal networks are a hallmark of percolation processes. In percolation, a system-spanning network is formed through the binding of building blocks. Percolation models have a long history in the study of polymer sol-gel transitions[48], and more recently, they have been proposed to play an essential role in biomolecular condensates[49,50]. Our observation of a fractal network within Bik1 droplets provides direct structural evidence for percolation in fluid biomolecular condensates. Based on these findings, we hypothesize that the fractal structure enables Bik1 dimers to achieve a high viscosity while maintaining a relatively low protein concentration in their condensed phase[19].

As a limitation of our study, we can only speculate about the implications of our in vitro findings obtained on a single protein for +TIP body behavior and function in budding yeast cells. One distinctive property of Bik1 is that it can phase separate into liquid droplets on its own, which contrasts with the other two core +TIP body components Kar9 and Bim1, which separately form aggregates or do not phase separate, respectively[14]. Bik1, and its C-terminal EEY/F-like motif in particular, thus seems to be a key component determining the material properties of the +TIP body as a whole. Whether the fractal structure of the Bik1 condensate is also present in the Kar9-Bim1-Bik1 condensate needs to be assessed in follow-up studies. However, if this were indeed to be the case then the achieved high viscosity at a relatively low protein concentration of +TIP body droplets could indeed offer the glue-like, mechanical-coupling properties to drive nuclear positioning during budding yeast mitosis and mating[6].

On the technical side, our study employed a combination of methods to gain molecular-level insights into the mechanism of protein phase separation. SAXS is a method of relatively low resolution that can examine the nanoscale conformation of biomolecules. By combining SAXS data with experimentally or computationally derived structures, this technique becomes a powerful tool for determining the shape of proteins in dilute solutions as well as their arrangement in condensed phases. XL-MS on the other hand defines protein-protein interactions at the single residue level and probes the conformational states of proteins in different solution conditions. By combining these approaches, we obtained a comprehensive view of the structure, dynamics, and interactions of Bik1, offering both global and local insight into the process of phase separation of a complex oligomeric and multidomain protein. We think that this powerful experimental framework can be extended to other biomolecular condensates, including more intricate +TIP networks.

In summary, our study provides a deeper understanding of conformational changes and mechanisms involved in the phase separation

of a complex multidomain, oligomeric protein. We present an experimental framework for investigating the supramolecular organization of phase-separated protein networks. In our model system, we found structural evidence supporting recent models emphasizing the role of percolation in biomolecular condensates. Gaining more and more insight into how proteins arrange themselves within complex condensed phases will ultimately enhance our understanding of the role of protein phase separation in fundamental cellular processes.

## Methods

### Bik1 dimer structure prediction
The structures of the Bik1 FL dimers shown in Figs. 1A, 2C, and Supplementary Fig. 1A were generated with AlphaFold[21] using ColabFold[51] v 1.2.0. with default settings. The input amino acid sequence is available in the Source Data file.

### Protein purification
The DNA encoding *S. cerevisiae* Bik1 (Uniprot ID: P11709) variants (mNG-Bik1 FL, Bik1 FL, Bik1 ΔN, Bik1 ΔQQFF, Bik1 ΔNC) were cloned into the pET-based bacterial expression vector PSPCm2, which encodes for the N-terminal 6×His-tag followed by a PreScission cleavage site using a positive selection method[52]. The Bik1 FL and mNG-Bik1 FL clones have already been described in our previous study[14]. All primers used in this study are summarized in Supplementary Table 4.

All protein production was performed in the *E. coli* strain BL21-CodonPlus (DE3)-RIPL (Agilent #230280) in an LB medium containing 50 μg/ml of kanamycin as described before[14]. Briefly, when the cultures reached an $OD_{600}$ of 0.6 at 37 °C, they were cooled down to 20 °C, induced with 0.75 mM isopropyl 1-thio-β-D-galactopyranoside (IPTG), and shaken for another 16 h at 20 °C. After harvesting by centrifugation, the cells were sonicated in the presence of protease inhibitors (cOmplete cocktail; Roche) and 0.1% bovine deoxyribonuclease I in purification buffer (20 mM Tris-HCl, pH 7.4, supplemented with 500 mM NaCl, 10 mM imidazole, and 2 mM β-mercaptoethanol).

Proteins were purified by immobilized metal-affinity chromatography (IMAC) on a HisTrap HP nickel-Sepharose column (GE Healthcare) at 4 °C following the manufacturer's instructions. The column was equilibrated in IMAC buffer A (20 mM Tris-HCl, pH 7.4, supplemented with 500 mM NaCl, 10 mM imidazole, and 2 mM β-mercaptoethanol). Proteins were eluted by IMAC buffer B (IMAC buffer A containing 400 mM imidazole in total). In the case of tag removed Bik1 FL (only for phase separation experiments), the N-terminal 6×His-tag was cleaved off by an in-house produced HRV 3C protease[53] in IMAC buffer A for 16 h at 4 °C. Then, samples were reapplied on the IMAC column to separate the 6×His-tag from the uncleaved protein.

Protein samples were concentrated and loaded onto a size-exclusion chromatography (SEC) HiLoad Superdex 200 16/60 (mNG-Bik1 FL, Bik1 FL, Bik1 ΔQQFF,) or Superdex 75 16/60 (Bik1 ΔN, Bik1 ΔNC) columns (GE Healthcare), which were equilibrated in SEC buffer (20 mM Tris-HCl, pH 7.5, supplemented with 500 mM NaCl, 10% glycerol, and 1 mM DTT) or in XL buffer (10 mM HEPES-NaOH, pH 7.5, supplemented with 500 mM NaCl and 1 mM DTT) for XL-MS experiments. In the case of Bik1 ΔN and Bik1 ΔNC, the SEC buffer contained 150 mM NaCl final concentration. Bik1 protein-containing fractions were pooled and concentrated to the desired concentration. The protein quality and identity were assessed by SDS-PAGE and mass spectrometry, respectively.

### Droplet imaging and generation of the phase diagram
Phase diagrams were constructed by assessing droplet presence under varying Bik1 FL and NaCl concentrations. The presence of droplets is assessed by diluting a Bik1 FL stock (final concentration 2.5–20 μM) into the buffer (20 mM Tris-HCl, pH 7.4) supplemented with NaCl to a final concentration between 150 and 500 mM on the well of a standard TC plate (Sarstedt). Upon mixing, each condition is immediately checked under a ZOE™ fluorescent imaging system (Bio-Rad) operated in bright field mode at 20× magnification. Each condition was assessed in three independent experimental replicates. Droplet screening of Bik1 FL and its truncated variants (10 mM HEPES-NaOH, pH 7.5) was performed in the 2.5–50 μM protein and 50–500 mM NaCl concentration range and samples were imaged on a Delta Vision microscope (Cytiva) operated in bright field mode at 100× magnification.

### Circular dichroism (CD) spectroscopy
CD spectrum of Bik1 FL (0.2 mg/ml in 20 mM Tris-HCl, pH 7.4, supplemented with 500 mM NaCl) was recorded at 25 °C on a Chirascan-Plus spectrophotometer (Applied Photophysics Ltd.) equipped with a computer-controlled Peltier element using a quartz cuvette of 1 mm optical path length. CD spectra were recorded between 200 and 260 nm. Thermal unfolding profiles were recorded by CD at 222 nm by continuous heating at 1 °C min⁻¹.

### Fluorescence microscopy
10% mNG-Bik1 FL/90% Bik1 FL was diluted in buffer (20 mM Tris-HCl, pH 7.4, supplemented with 250 mM NaCl) to a final concentration of 3.4 mg/ml. The phase-separated droplets were deposited onto glass coverslips coated with PEG-Silane (Gelest) and sealed with a second coverslip with a spacer in between. Samples were imaged on an inverted confocal microscope (Nikon Ti2 Eclipse with Yokogawa CSU-W1 spinning disc) using a 60× water immersion objective (NA of 1.2).

For quantification of intensities inside droplets, acquired z-stacks were analyzed with a MATLAB code. Briefly, droplets were located above a threshold intensity in 3D, and the center z-plane for each drop was identified. In each mid-plane, the centroid of the droplet circle was identified, and intensities averaged azimuthally as a function of distance from the droplet surface. Radial intensities of 233 different droplets were then averaged to give a representative profile of mNeonGreen intensity as a function of distance from a droplet edge.

### Crosslinking mass spectrometry (XL-MS)
Crosslinking reaction: Different Bik1 samples were subjected to a crosslinking reaction with 1 mM isotope labeled disuccimidylsuberate[35] (DSS-d0 and DSS-d12; CreativeMolecules Inc.). Crosslinking experiments with DSS for both qualitative and quantitative XL-MS were performed in three independent replicates. For the crosslinking experiments of Bik1 in the pellet and supernatant fractions, Bik1 was diluted into 50 μl of 150 mM NaCl XL-MS buffer and was centrifuged for 15 min at 4 °C, 16,000 × g. Prior to the crosslinking reaction, the pellet was resuspended in 50 μl of 150 mM NaCl XL-MS buffer. The reaction of Bik1 variants with the acid crosslinking reagent was initiated by the addition of a mixture consisting of pimelic acid dihydrazide (light and heavily labeled: PDH-d0, PDH-d10) at 8.9 mg/ml with DMTMM at 12 mg/ml. Crosslinking experiments with PDH were performed without replicates. The reaction was quenched by removing the reagents with a Zeba Spin Desalting column[36] (0.5 mL, 7 K MWCO, Pierce). Upon quenching, samples were dried, dissolved in 8 M Urea, reduced with 5 mM TCEP, and alkylated with 10 mM iodoacetamide. The proteolysis was performed overnight at 37 °C in 1 M urea using different proteases (trypsin 1:50 E:S; AspN 1:200 E:S; AspN/GluC 1:100 E:S; GluC 1:100 E:S; chymotrypsin 1:100 E:S) and then quenched with 5% formic acid. Generated peptides were subjected to cleanup (C18 column, TheNest Group) and separated by SEC using an ÄKTA micro chromatography system (GE Healthcare) using a Superdex 200 Increase 3.2/30 column (Cytiva). The SEC fractions were then dried and re-dissolved in 5% acetonitrile and 0.1% formic acid for mass-spectrometry analysis. For the targeted analysis of crosslinked peptides, samples were not subjected to SEC fractionation.

Data acquisition: Liquid chromatography-tandem mass spectrometry (LC-MS/MS) measurements were performed on two different LC-MS systems. The qualitative crosslinking analysis was performed in

Data Dependent Acquisition mode (DDA) on an Orbitrap QExactive+ MS coupled to an EASY-nLC-1000 liquid chromatography system. Peptides were separated using a reverse phase column (75 μm ID × 400 mm New Objective, in-house packed with ReproSil Gold 120 C18, 1.9 μm, Dr. Maisch GmbH). The quantitative crosslinking analysis was performed in Parallel Reaction Monitoring mode (PRM) on an Orbitrap Qexactive+ MS coupled to a Dionex Ultimate 3000 RSLCnano System liquid chromatography system. Peptides were loaded on a commercial trap column (μ-Precolumn C18 PepMap100, C18, 300 μm I.D., 5 μm particle size) and separated using a reverse phase column Thermo PepMap100 C18 (50 cm length, 75 μm inner diameter, 35 μm particle size). For the identification of crosslinking peptides in DDA mode, an LC method was set up to separate peptides across a 60 min gradient: from 5% to 25% in 55 min and from 25% to 40% in 5 min (buffer A: 0.1% (v/v) formic acid; buffer B: 0.1% (v/v) formic acid, 95% (v/v) acetonitrile). The acquisition method was performed with one MS1 scan followed by a maximum of 20 scans for the top 20 most intense peptides (TOP20) with MS1 scans ($R = 70{,}000$ at 400 m/z, maxIT = 64 ms AGC=1e5), HCD fragmentation (NCE = 25 or 28%), isolation windows (1.5 m/z) and MS2 scans ($R = 35{,}000$ at 400 m/z, maxIT = 110 ms, AGC = 5e4ms). A dynamic exclusion of 30 s was applied, and charge states lower than three and higher than seven were rejected for the isolation.

For the targeted quantification of crosslinking peptides in P.R.M. mode, an LC method was set up to separate peptides across 45 min gradient: from 8% to 25% in 40 min and from 25% to 50% in 5 min (buffer A: 0.1% (v/v) formic acid; buffer B: 0.1% (v/v) formic acid, 80% (v/v) acetonitrile). The MS acquisition of targeted peptide (17 crosslinks heavy and light), 1 householder peptides for Bik1 peptides was set up with the combination of one MS1 untargeted scan ($R = 70{,}000$ at 400 m/z, maxIT = 64 ms, AGC = 3e6) and 37 scheduled targeted scan ($R = 70{,}000$ at 400 m/z, maxIT = auto, AGC = 1e5) using an isolation window of 1.4 m/z and HCD fragmentation (NCE = 28%).

Peptide identification and data analysis: For the identification of crosslinking data, mass spectra were converted to mzXML format and searched with xQuest/xProphet[54] against a database containing the FASTA sequence of Bik1 and its decoy sequence. DSS Crosslinked peptides with ld.Score >20 and filtered for consistency (identification in at least two out of three independent experimental replicates) were considered in the analysis. PDH Crosslinked peptides with ld.Score >25, deltaS >0.9, and FDR < 0.05 were considered in the analysis. XL peptides were visualized using the xiVIEW web-based tool for the analysis of crosslinking results[55]. Conformo-specific crosslinked peptides (crosslinks identified only in specific conditions) were selected and analyzed using a targeted proteomics approach[40,56]. We generated a library based on the previous xQuest identification; only common transitions from light and heavy crosslinked peptides were used for this analysis. The quantification of targeted peptides was manually performed using Skyline-daily[57] based on the following criteria: (1) co-elution of heavy and light crosslinked peptides, (2) matching of the peak shape, and (3) matching intensity for at least five common transitions of heavy and light crosslinked peptides (transitions shared between the light and heavy form of the crosslinked peptide). The abundance of crosslinked peptides was then calculated by summing the integrated area of ten common transitions per peptide (five transitions for the heavy and light forms of the crosslinked peptide, respectively). Crosslinked peptides were normalized for the intensity of a non-crosslinked Bik1 peptide (MVLEEVQPTFDR). The significance of change for the log2 abundance of the crosslinked peptides was estimated with $p$-values using a two-sided, not paired $t$-test.

To account for potential artifacts from crosslinking reagent concentration and detection based on different collision energy during fragmentation, we performed the crosslinking reaction of Bik1 FL in solution at different DSS concentrations (0.5 mM and 1 mM) and distinct fragmentation settings (NCE = 25 and 28). We identified 117

crosslinked peptides including 27 monolinks using a filtering cutoff of ld.Score >20 and FDR < 0.05. We consistently identified 19 crosslinking peptides across all experimental conditions tested. Notably, we observed a higher degree of overlap in the results when conducting the crosslinking reaction at different concentrations while maintaining the same fragmentation energy, as compared to varying the fragmentation energy while keeping the concentration of the crosslinking reagent constant. As increasing the concentration of crosslinking reagents from 0.5 mM to 1 mM did not result in more self-crosslinked peptides, we proceeded to study differences in interactions between dilute and phase-separated Bik1 FL using 1 mM DSS.

The entire dataset, including raw data, generated tables, and scripts used for the data analysis are available in the PRIDE repository[58]. PXD050928, effect of DSS concentration and fragmentation on the identification of XL peptides (Supplementary Data File 2); PXD050929, identification of DSS XL peptides in Bik1 variants and different conditions (Supplementary Data File 1); PXD050930, identification of PDH XL peptides in Bik1 at different conditions (Supplementary Data File 1); PXD055703, quantification of DSS XL peptides in Bik1 variants and different conditions (Supplementary Data File 3).

## Size-exclusion chromatography followed by small-angle X-ray scattering (SEC-SAXS)

SEC-SAXS measurements were performed at beamline B21[59] at the Diamond Light Source using the photon energy of 13.1 keV. A SEC step was performed for each sample prior to the SAXS measurement using a Shodex KW-404 column equilibrated in SAXS buffer (20 mM Tris-HCl, pH 7.5, supplemented with 500 mM NaCl, 2% glycerol, and 1 mM DTT) and connected in-line to the X-ray scattering measurement cell. Samples of Bik1 FL, Bik1 ΔQQFF, and Bik1 ΔNC were injected at concentrations of 4, 6, and 8 mg/ml in 50 μl volumes and with a flow rate of 0.16 ml/min.

The ATSAS[60] software was used to perform buffer subtraction, SEC peak scattering intensity summation, radius of gyration (Rg) calculation from Guinier plots, and analysis of distance distribution functions P(r). Ab initio calculation of molecular envelopes was performed using DAMMIF[61]. Model averaging and pairwise cross-correlation were performed using DAMAVER[62]. For all analyzed proteins, 20 bead models were calculated using random start seeds with no presumed internal symmetry (P1). For the final envelope, the most similar 19 models (out of 20) were averaged for each protein. The resolution of the calculated envelopes was estimated using SASRES[63]. Graphic visualization of protein molecules was done using PyMOL (the PyMOL Molecular Graphics System, Version 2.4.1 Schrödinger, LLC).

## Small-angle X-ray scattering (SAXS) of phase-separated Bik1 samples

Data collection: SAXS experiments were conducted at the cSAXS beamline at the Swiss Light Source (Paul Scherrer Institut). The monochromatic beam, with a photon energy of 11.2 keV, corresponding to a wavelength of 0.11 nm, was focused so that the full width at half maximum (FWHM) beam size at the sample position was $12 \times 26$ μm along and across the flow direction in the microfluidic channel (Fluidic 394, featuring a $200 \times 200$ μm channel, #10000016 Microfluidic ChipShop), respectively. This beam size was chosen to ensure the entire beam fits into the microfluidic channel without any edge scattering.

A microfluidic chip with a sample continuously flowing through the volume illuminated by X-rays was used. Therefore, the sample in the beam was continuously renewed with fresh material, and any aging effects or degradation of the sample due to the X-rays can be ruled out. We estimated the number of droplets passing the beam in the time frame of one second. The volume exposed to the beam was $12 \times 26 \times 200$ μm³ (12 μm along the direction of the flow, 26 μm perpendicular to the direction of the flow and the X-ray beam, and 200 μm

along the X-ray beam), and the volume fraction of micron-sized droplets was approximately 0.003, which means that ~40 droplets were in the beam at any instant. The sample was pumped through the capillary with a velocity of 7.5 mm/s, which resulted in ~3 × 10⁴ droplets passing the beam per second. The sample is renewed 625 times in an exposure of one second. Each SAXS measurement presented in this study corresponds to 40 exposures of one second that were averaged, which means that each measurement results from ~10⁶ droplets passing the X-ray beam.

Scattered X-rays were detected using a Pilatus detector[64], positioned 2.172 m downstream of the sample, as determined by analyzing the scattering signal of a silver behenate standard. The flight path between the sample and detector was evacuated to minimize parasitic scattering and absorption. A beamstop served both to protect the pixel detector from the direct beam and to allow for measuring the intensity of the unscattered beam.

To mitigate radiation damage, pre-formed droplets of Bik1 FL (4 mg/ml) in 20 mM Tris-HCl, pH 7.4, buffer supplemented with 260 mM NaCl were introduced into a microfluidic chip at a pumping rate of 0.1 μl/s. To minimize sample consumption while maintaining high sample flow rates, the droplet solution was co-flowed with 2 × 0.1 μl/s of buffer, which was fed through the left and right channels in the microfluidic cross junction, leading to a sheet flow with a dilution of the sample along the channel. A few seconds after the channel was filled with the sample, data were acquired at 1.5 cm, 2 cm, and 4.5 cm from the mixing point, respectively. The corresponding signal from the buffer was measured at approximately the same distance from the mixing point. This buffer signal was subtracted from the signal of the protein-buffer sample to generate the protein scattering signal reported in the manuscript. In principle, this protein scattering signal could have contributions from protein in the dense phase, protein in the dilute phase, protein at droplet interfaces, and from the overall droplet shape. From our previous measurements of the phase behavior of Bik1 droplets[14], we know <7% of the protein is in the dilute phase. From the dimensions of the droplets (~10 μm) and protein (~10 nm) we know that <0.1% of the protein lies at the interface between the dense and dilute phases. Contributions from scattering of the droplet as a whole are not significant in this q-range, which is only sensitive to droplets with radii less than 300 nm. Therefore, the buffered-subtracted scattering signal in these experiments is dominated by scattering from the interior of droplets[65].

Data fitting with Gaussian blobs: To estimate the mass distribution on the length scale of many Bik1 dimers, we fit the SAXS data using 4000 Gaussian blobs with a standard deviation $\sigma$ of 40 Å. The blobs are arranged in a spherical volume with a diameter of 10,000 Å, which is large enough to cover the largest length scale that can be resolved by our SAXS measurements. The use of Gaussian blobs for our data fitting approach is convenient as the real space and the reciprocal space representations are described by Gaussians. This approach provides a possible configuration that is orientationally averaged for the fit to the SAXS data, as many droplets contributed to a measurement. Therefore, the scattering curve for the fit is calculated as the average resulting from 300 random orientations of the momentum transfer vector. Least-square fits to the SAXS data were performed using a self-developed code written in the Julia language[66].

The form factor of the sphere containing the Gaussian blobs must be suppressed, as it would disturb the fit at low $q$. This is achieved by using a Gaussian window placed at the center of mass of the set of blobs used to fit the measurement and decaying from the origin to the outward. The scattering amplitudes of all blobs in the volume are multiplied with this window function, which corresponds to a convolution with a Gaussian in reciprocal space. The window function is used with a standard deviation $\sigma_w = 3000$ Å, which sets the $q$ resolution of the fit to $2\pi/\sigma_w \approx 0.002$ Å⁻¹.

Data fitting with spherocylinders: Our SEC-SAXS results revealed an elongated, cylindrical conformation for the Bik1 dimer, which is considerably different than Gaussian blobs. They further suggested that the overall conformation of the protein is dominated by its two-stranded coiled coil with little if any, contributions from its N- and C-terminal domains. Based on our computational analysis, this coiled coil can be divided into three segments that are connected by flexible linkers. To fit our SAXS data with a more realistic representation of the Bik1 dimer, we thus constructed three models with increasing complexity: model 1 consists of a single spherocylinder with a radius of 12 Å and a length of ≈300 Å; model 2 consists of two linked spherocylinders with radii of 12 Å and lengths of 155 Å and 135 Å, respectively; model 3 consists of three linked spherocylinders with radii of 12 Å and lengths of 155 Å, 76 Å, and 50 Å, respectively.

To fit the SAXS data using the three models, we placed 50,000 or 100,000 model copies according to the mass distribution obtained from the Gaussian-blob fit in a spherical volume with a diameter of 6000 Å. This chosen diameter restricts our analysis to $q > 0.004$ Å⁻¹ or a maximum distance of about 1500 Å, which is much larger than the length of a fully elongated Bik1 dimer and is therefore acceptable. Due to the reduced volume, the window function is used with $\sigma_w = 1500$ Å and, therefore, the $q$ resolution of the fit is reduced to $2\pi/\sigma_w \approx 0.004$ Å⁻¹, which does not limit our analysis of the SAXS data. All least-square fits to the SAXS data were performed using Julia code[66] for the calculation of the scattering signals resulting from spherocylinders as well as for the detection of spherocylinder clashes in the volume used to fit measurements.

For fits using model 1, spherocylinders with random orientations are placed close to the centers of the Gaussian blobs before translations, and reorientations of them are carried out to improve the fit to the SAXS data. While translations and reorientations are chosen randomly, only those improving $\chi^2$ of the fit are accepted, while all others are rejected. Good fits were obtained for all three SAXS measurements. This result was achieved with two volume fractions, $\varphi \approx 0.05$ and $\varphi \approx 0.1$, for all three SAXS measurements. Model 2 approximates a Bik1 dimer with two spherocylinders with a flexible angle $\alpha$ between the cylinder axes. In the stretched-out configuration ($\alpha = 180°$), model 2 mimics model 1, while the two spherocylinder components lie parallel to each other at $\alpha = 0°$ in the fully folded-back configuration. The result obtained with model 1 was used to start the data fitting using model 2. The longer of the two segments of model 2 was superimposed on one end of each copy of model 1, and the second segment was added with a random orientation in a manner to avoid clashes with neighboring model copies. The second segment was placed by 'rolling' its spherical cap on the cap of the first segment until the desired orientation was reached. The fit to the SAXS data was performed with random translations of model 2 segment pairs and random reorientations of the first or second segment of the pair, always maintaining the link of the two segments and avoiding clashes with neighboring model copies. Analogous to the transition from model 1 to model 2, the data fitting with model 3 was started using the result obtained with model 2. Good fits to the data were obtained, and the orientation of the three model segments was close to random with a slight tendency towards a fully stretched-out configuration for both angles $\alpha$ and $\beta$ between model segments. For the minimization of $\chi^2$, the same procedure using reorientations and translations is used for all three models. The fits to the data obtained with models 1, 2, and 3 were of comparable high quality.

The local nematic order parameter, $S$, was determined from the model 1 fits obtained with 50,000 or 100,000 model-1 copies. The model-1 copies in cubic boxes with side lengths 400 Å were considered to calculate the local order parameter tensor. The local order parameter is obtained as $S = 3/2*e$, where $e$ is the largest positive eigenvalue of the order parameter tensor. With boxes containing 30 or more model-1 copies, the order parameter is below 0.4, which we interpret

as an orientationally unordered configuration. With 50,000 copies, the order parameter can be higher than 0.4 when $N < 30$ and the spherocylinders happen to be quite well aligned.

## Reporting summary

Further information on research design is available in the Nature Portfolio Reporting Summary linked to this article.

## Data availability

Custom R script code and MS files generated in this study have been deposited in the PRIDE archive, a public data repository for MS proteomics data (https://www.ebi.ac.uk/pride/), under the following accession codes: PXD050928, PXD050929, PXD050930, PXD055703. The SEC-SAXS datasets are available in the SASDB archive, Small Angle Scattering Biological Data Bank (https://www.sasbdb.org/) with the following accession codes: SASDUT6, SASDUU6, SASDUV6. The SAXS datasets collected for the phase-separated droplets of Bik1 have been deposited in the Zenodo (https://zenodo.org/) repository, with the following https://doi.org/10.5281/zenodo.10944717. Source data are provided as a Source Data file. Source data are provided with this paper.

## Code availability

The code used to fit the SAXS curves of phase-separated droplets of Bik1 can be found in the Zenodo repository with the following https://doi.org/10.5281/zenodo.13822678.

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

## Acknowledgements

We thank Sandro Meier (ETH Zürich) for the purification of mNG-Bik1 FL, the staff of the DLS beamline B21 of the Diamond Light Source for the provision of and excellent assistance with SAXS beamtime. We are grateful to Alexander Leitner (ETH Zürich) and Carla Schmidt (JGU University of Mainz) for their insights into crosslinking strategies and for access to instruments and data analysis platforms. We acknowledge financial support from the Swiss National Science Foundation Sinergia grant CRSII5_189940 (to E.R.D. and M.O.S.). M.P.C. has received funding from the European Union's Horizon 2020 research and innovation program under the Marie Skłodowska-Curie grant agreement No 884104 (PSI-FELLOW-III-3i). C.L. has received funding from the Deutsche Forschungsgemeinschaft (Proj.- Nr. 523842861, LO 3088/1–1). This work benefited from access to Beamline B21 at Diamond Light Source and has been supported by iNEXT-Discovery, project number 871037, funded by the Horizon 2020 program of the European Commission.

## Author contributions

M.P.C. and M.O.S. designed the project and wrote the first draft of the manuscript. M.P.C. and T.G. cloned and purified proteins and performed phase separation studies. M.P.C., T.G., and F.U. designed, performed, and analyzed crosslinking experiments. K.R. performed and analyzed confocal microscopy experiments. M.P.C. and I.V. prepared and analyzed SEC-SAXS measurements. M.P.C., T.G., C.P., A.M., and U.G. designed and performed SAXS measurements of phase-separated Bik1 samples, U.G. analyzed and modeled the obtained results, and E.R.D. and C.M.L. supported data analysis. M.O.S. supervised the entire study. All authors contributed to the writing and data interpretation of the manuscript.

## Competing interests

The authors declare no competing interests.
