## [Transparent Peer Review file · Nature Communications]

Phase separation of a microtubule plus-end tracking protein into a fluid fractal network

Corresponding Author: Professor Michel Steinmetz

Version 0:

Reviewer comments:

Reviewer #1

(Remarks to the Author)

Phase separation of a microtubule plus-end tracking protein into a fluid fractal network.
Czub et al.

The submitted manuscript aims to dissect the structural details underpinning the phase separation of the Bik1 protein. In this reviewer's opinion, the Kar9 network is one of the more exciting systems proposed to have a phase-separation mechanism. As such, ascertaining these details is essential in understanding how these cellular processes work. The manuscript contains three themes: 1) understanding the structure of dilute phase Bik1, 2) Uncovering peptides within Bik1 that form crosslinks, and 3) Examining the mesoscale structure of Bik1 droplets. In brief, the research methods presented here are sound; the analysis generally follows the experimental results with little over-speculation, and everything is explained in an easy-to-follow narrative. In addition, the authors show the value of two underused methods (quantitative mass spectrometry and SAXS) in building a coherent, structural picture of phase separation from the single molecule to the mesoscale. Following are my comments.

I appreciate the authors' use of AF structures as the starting point and their avoidance of over-interpretation. It needs to be made clear how the phase diagram in Fig 1D was built. I suspect that a series of samples were prepared and assessed for droplets. Perhaps I missed this, but the methods involved in this assay should be quite detailed as they are notoriously sensitive.

The data and schematic in Figure S4 should be in the main text. The data is helpful in demonstrating that homogeneity is a function of length scale, and the schematic helps convey potential structures. That said, the schematic would be a bit more useful if drawn to reflect the mass spec results.

I am typically very suspicious of DAMMIF (or similar) modeling of SAXS data. However, in this case, it seems to make sense. If it is easily accessible, I would like to see if the models fit to the raw data. My personal preference is also to see SAXS data on flexible systems presented in log-log as opposed to semi log (as the authors show in the later figures with phase separated samples). This helps accentuate the scaling in the data. While the DAMMIF models seem reasonable, I could argue that I could come to the same conclusion simply by looking at the $P(r)$ plot combined with the rod-like scaling combined with some flexibility as a function of the length scale shown in the Kratky plot.

It is unclear how sensitive the mass spec analysis is. I'm also not sure how much sample is being measured. What I would like to know is how the thresholding for identifying a peptide relates to the likelihood of that crosslink existing. Does the crosslink need to exist in 50% of Bik1 molecules or would you detect it if it only occurred in 1% or less?

I appreciated the quantitative approach to the mass spec analysis presented in Figure 4. A more schematic view of how the structure evolves between dilute and phase-separated proteins would benefit the manuscript.

It was quite interesting to me that the dilute (supernatant) sample did not clearly cluster with any of the other conditions (and seemed heterogeneous between samples). The dendrogram indicates that the supernatant is closer to the phase-separated / pellet samples than the rest. One would expect that the low ionic strength condition would cause the dimer to form some small oligomers and perhaps be missing some long-distance contacts that are sterically impossible in these smaller structures. However, these would be distinct from the high salt conformations. It would be good to see a more clear differential between these states.

I have little to nothing to say about the condensed state SAXS data other than that it is delightful to see data that looks this clean and is a model that makes logical sense and fits well. I think that the radial distribution function that came from the models is something that many SAXS experts could have drawn by eye looking at the raw data. What is interesting to me is

that the author's cylinder models fit the data well to wider angles. Given the technical nature of this manuscript, I think it would be useful to have more discussion of these features in the main text. It would also be useful to see things like the order parameters and angle distribution not buried in the supplemental information.

In my experience, this method of measuring SAXS data for condensed phase samples shows some heterogeneity due to the volume fraction of droplets within the beam in a given integration time. Was the integration time long enough, or were the droplets sufficiently small relative to the beam size to avoid this issue? If not, it would be good to show some of the individual measurements prior to averaging. If so, it might be useful to mention in the methods.

Final minor note: the manuscript mentions the availability of the code for analyzing the mass spec data. This should include the SAXS data as well. The methods were a bit high-level with respect to how the analysis and fitting were done.

Reviewer #2

(Remarks to the Author)

In their study Czub et al. investigate the role of the microtubule plus-end tracking protein Bik1 during phase separation. The authors express different variants of Bik1 and study their behavior in in-vitro phase separation assays by cross-linking mass spectrometry (XL-MS) and small-angle X-ray scattering (SEC-SAXS) to investigate domain organization and conformation of Bik1 and potential changes therein during phase separation. Studying the internal organization of proteins and biomolecules during phase separation is an important topic and both XL-MS and SEC-SACS are well suited techniques to experimentally address this problem.

In their manuscript the authors show an AlphaFold prediction of Bik1 together with a phase diagram of its condensation behaviour and a sketch of a number of Bik1 variants (Figure 1). However, only a subset of these variants are experimentally followed-up by SAXS (Figure 2 and 5) and only one variant is fully examined by qualitative (Figure 3) and quantitative (Figure 4) XL-MS in vitro. Here, the authors perform two sets of cross-linking experiments. In the first one, they cross-link Bik1 under high salt (dilute) and low-salt (phase separating) conditions in bulk using two complementary cross-linkers (DSS and PDH). In the second set of experiments they follow a subset of DSS induced cross-links, which they term "conformospecific" as they are only identified in either of the states, in a quantitative manner using a targeted proteomics approach.

Overall, these experiments are technically sound and carried out state of the art (of note the reviewer is no expert in SEC-SACS and therefore focuses mainly on the mass spectrometry part of the study). However, both experimental set-ups have been described previously in detail, also in the context of phase separating proteins. Moreover, the conclusions that the authors draw from the data appear in their current form overstated and require additional experiments to be supported by the data. In its current form the manuscript appears therefore too premature for publication in Nature communications and likely better suited for a more focussed journal.

Major concerns that need to be addressed.

1. Qualitative XL-MS (Figure 3, Figure S6).

Cross-links in full-length Bik1 are compared in low salt vs high salt conditions (Figure 3). Major conclusions on domain structure, flexibility, influence of multivalency and protein networks of Bik1 in condensates are drawn from differences in single cross-links. As powerful as XL-MS is as an approach, this should not be done without additional experiments and/or follow-up studies that confirm the conclusions, as any cross-linking information contains a certain amount of variation in itself and this variation tends to be larger within condensates and when working with proteins that contain intrinsically disordered domains.

a) The authors need to repeat these experiments using the other Bik1 constructs shown in Figure 1 - full length (FL), Δ CG, Δ tail, CC and Δ QQFF. Only by using these variants that miss domains and/or intrinsically disordered parts of Bik1, will the authors be able to confidently draw conclusions from changes in cross-linking patterns on phase separation behaviour.

b) Their approach contains a mix of Bik1 in condensates and in the dilute phase (as a significant portion of Bik1 under low salt conditions will not partition into condensates). The authors therefore need to compare their data to the pellet and supernatant fraction in Figure S6, which should also consist of Bik1 in the condensed and dilute phase. However, even from a quick glance at the data it is obvious that there are large discrepancies between the datasets that should contain the same sample type - i.e. 3A (dilute) and S6 (supernatant) or the condensed forms 3A (phase separated) and S6 (pellet). The authors need to explain this discrepancy.

c) Δ QQFF Bik1 is probably the most interesting variant of the study - as four missing amino acids preclude phase separation. The authors should make further sub-constructs and also test different salt conditions under which Δ QQFF phase separates.

d) In order to draw robust conclusions on protein-protein contacts - as for example insinuated by the title of the manuscript - the authors should not only use "selflinks" but need to repeat these experiments by mixing labelled and unlabelled forms of Bik1

2. Quantitative XL-MS data (Figure 4).

a) Also in this case the experiments need to be repeated using additional Bik1 variants, see above. Critically, conditions under which Δ QQFF starts to phase separate should be investigated and included.

b) The comparison of the existing full-length data itself appears, at least partly, inconclusive.

Why do cross-links within/between the same domains show opposite behaviour (Figure 4D)?

For example – why are the links 7-359 and 16-338 down regulated but 23-374 or 79-338 upregulated?

(This must not necessarily be a problem. However, if the authors, as they do in the current form of the manuscript, draw conclusions from changes in single cross-links during phase separation, then the authors need to explain).

c) As above, the authors need to explain why differences between samples that should contain the same state (i.e. dilute/supernatant vs pellet/phase sep) are at least partly in larger disagreement than samples that should contain different states (Figure 4C).

d) Figure 4 panels 4C and D are quite confusing. Similar links should be mapped onto the structure or the 2D model – as in Figure 3 – and up- and down regulated links should be indicated.

3. Comparison between SAXS and XL-MS data.

There is very little comparison between the datasets.

Minor Comments

Was the filtering strategy only based on spectra quality (Id.Score > 25 and FDR < 0.05), as written in the methods, or also on consistency (crosslinked peptides must be identified in at least two out of three replicates) – see page 8 of the manuscript? This is quite an important information and should be clarified.

The colors in Figure 4D are very hard to see.

The title appears to be an overstatement and should be changed.

Reviewer #3

(Remarks to the Author)

This study investigates the phase separation and internal organization of Bik1, a microtubule plus-end tracking protein (+TIP) from the CLIP-170 family, which plays a crucial role in budding yeast cell division. The authors demonstrate that Bik1 exists as a rod-shaped dimer, with its conformation predominantly influenced by a central coiled-coil domain. During the process of liquid condensation, Bik1 undergoes significant conformational rearrangements, resulting in a 2-3-fold increase in interactions between the protein's folded and disordered domains. This highlights a complex interplay between different structural regions of the protein. This paper also suggests that the supramolecular structure of Bik1 condensates is not homogeneous like classical liquids, but rather displays a heterogeneous, fractal architecture characterized by distinct protein-rich and protein-free domains. The detailed insights from this study shed light on the physical properties and dynamic rearrangement of the Bik1 protein in both its dilute and condensed phases.

Major Comment

In recent years, +TIPs have garnered considerable attention due to their role in several microtubule-based cellular processes, with a recent proposition that they function as liquid condensates forming dynamic, membrane-less organelles. However, the detailed mechanisms governing the formation and internal organization of these condensates are yet to be answered as previous studies were primarily focused on the functional aspects of +TIP liquid-like condensates in microtubule dynamics. The experimental approach used here can be applied to other +TIP networks, potentially revealing new aspects of the organizational principles underlying microtubule-associated proteins.

While these techniques hold promise for robust structural characterization of proteins within condensates, this work is limited by its description of just a single component of the +TIP body, whereas it is already known that there are several other protein components such as Kar9, Bim1, among others, undergoing multivalent network interactions resulting in the formation of the liquid-like +TIP condensate. (as also mentioned by the authors). Currently, the field is moving towards more physiological reconstitutions, and reconstituting a more physiologically relevant +TIP body and then assigning these interactions and studying the change in the conformational landscape of each of proteins would be more relevant and informative towards understanding its role in modulating microtubule dynamics. If they want to stick to one protein, it's important to add this caveat under a limitations of the study section, and not overclaim on the relevance.

Minor Comments

1. "These fluorescence microscopy experiments suggest that the distribution of Bik1 dimers is uniform throughout the droplet; the fluorescence of Bik1 is over 60 times brighter in the condensed than in the dilute phase (Figures 1Figure 1E and S4A)."

Did the authors perform control experiments to ascertain that signal saturation did not cause underestimation of protein concentrations using fluorescence-based measurements.

2. The authors provide a phase diagram for the condensation of Bik-1 protein as a function of salt concentration where the protein seems to exhibit a threshold concentration of around 5 μ M at 250 mM NaCl (the concentration of salt chosen for experiments in this work).

Can the authors comment on what is the physiological concentration of this protein and how it relates to this condensation observed at μ M range of protein concentrations. Also, how does the phase behaviour compare to what was reported earlier for Bik1 in the Multivalency ensures persistence of a +TIP body at specialized microtubule ends | Nature Cell Biology (Maier et al. Nat. Cell Biol. 2023). (Ref 14 in the manuscript).

3. "Deletion of the CAP-Gly or the zinc-finger-containing domain (Bik1 Δ CG and Bik1 Δ tail, respectively) both suppressed protein condensation at low salt conditions (e.g., 250 mM sodium chloride) where Bik1 FL readily formed droplets. Therefore, we investigated whether mixing the two truncated variants under low salt conditions can rescue their potential to undergo condensation. 5 However, no visible droplets were observed under the same conditions used for Bik1 FL (Figure S3).....This finding indicates that the introduced truncations in both Bik1 variants reduced the multivalency of the system needed for efficient phase separation of Bik1 FL. Additionally, we examined the relevance of the C-terminal EEY/F-like motif QQFF of Bik1 for the protein's ability to undergo condensation. Interestingly, the deletion of the motif in Bik1 FL (Bik1 Δ QQFF; Figure 1C) was sufficient to prevent phase separation of this minimally truncated Bik1 variant (Figure S3)."

Have the authors tried to construct phase diagrams, like the WT Bik1, for the rest of the mutants? Is there a possibility that these mutants could still form condensates at lower salt concentrations (< than 250 mM NaCl) as the FL has several regions of positive and negative charges. Also, the authors have used a 250 mM NaCl solution for comparison of condensate formation propensity but have resorted to using a lower salt concentration in their cross-linking mass spectrometry experiments. Is there any reason for that?

Also, as the authors already highlight how the deletion of just 4 C-terminal residues (construct Δ QQFF) prevents condensation completely at the experimental conditions used in this work, can the authors comment on what might be the cause. Interestingly, the authors mention on Page 11 Lines 361-364, that the relative abundance of specific crosslinks for the Bik1 Δ QQFF were found to be somewhere in between the pelleted phase-separated fraction and the dilute dispersed fraction of WT Bik1, thereby, suggesting that formation of certain higher order oligomers in principle prevented the condensation of this construct. It would be interesting if the authors could characterize these oligomeric intermediates (if formed) and then how they prevent protein condensation.

4. "Four out of five selflink peptides were identified in the coiled-coil domain (Lys207, Lys245, Lys263, and Lys374)"
Were these interactions between the CC domain residues found in both the dispersed and condensed phases or are there any changes in this distribution to accommodate interactions with the CAP-Gly domain in the condensed phase? Also, can the authors speculate or show how the condensation behaviour might be affected by deleting or modifying sections of the CC domain either within the heptad repeats or the disordered linker sequences?

5. "Monolinks with reduced abundance levels in the pellet sample (Lys310 and Lys373). They can serve as proxies for assessing residue accessibility. This result suggests that in the Bik1 pellet condition, the coiled-coil region around Lys310 and Lys373 is much less exposed to the solvent than in other conditions. Differences in the coiled-coil residue exposure to the solvent or other Bik1 domains might be related to the presence of the kinks between coiled-coil segments,"
Can the authors comment on how the effective solvent property within the condensed phase might favour either of the Bik1 conformations: "rod-like or V-shaped". Apart from using this technique of cross-linking mass spectrometry, did the authors ever consider investigating the shift in the conformational paradigm of Bik1 in transitioning from the dispersed to the condensed phase, using site-specific fluorescence, NMR, and/or EPR-based measurements?

6. It is interesting to note the formation of a fractal-like network within the Bik1 condensates but can the authors comment on what is the exact significance of this sort of a network in the context of +TIP body and what could be its role in modulating microtubule dynamics. In particular, given that this is a single protein, the authors should have a paragraph on limitations of the study. The authors can expand on this and highlight its relevance in the biological context.

Version 1:

Reviewer comments:

Reviewer #1

(Remarks to the Author)

The revised manuscript has addressed all of my suggestions. I recommend it for publication.

Reviewer #2

(Remarks to the Author)

Reviewer 2: The authors have significantly improved the quality of their manuscript by adding new data and addressing important issues raised by the reviewers. In particular, by adding an additional quantitative XL-MS dataset for conform-specific peptides in a panel of Bik1 variants (Bik1 FL, Bik1 N, Bik1 NC, and Bik1 QQFF) at high and low sodium chloride concentration the authors have clearly strengthened their manuscript.

Unfortunately, fundamental points, which are central for their line of argumentation have not been addressed and thus the manuscript still requires additional revision before it can be accepted for publication.

Specifically, the cross-linking data is based on single (biological) measurements, as only technical replicates have been conducted. This is true for both the qualitative as the quantitative analysis (see Figure 3 and 4, in particular, Figure 3A and Figure 4B, C and D).

While for the qualitative analysis one may argue that by employing different linkers a somewhat comprehensive picture has emerged, this remains a central flaw of the quantitative data.

As argued before, the authors need to include true (biological) replicates, both to identify their list of conform-specific crosslinks (i.e. Figure 4A and 4B) AND then to conduct their quantitative analysis for their subset of conform-specific crosslinks for the panel of Bik1 variants (Figure 4D).

Data from mere technical replicates is not sufficient to allow for any robust interpretation of changes in crosslink abundances and thus protein dynamics, in particular for such dynamic systems as phase separated proteins.

As a side note. It is very hard to follow the revisions of the authors as no version of the revised manuscript with tracked-changes has been submitted.

Reviewer #4

(Remarks to the Author)

Reviewer #3 had one major comment concerning the focus on a single protein (Bik1) when +TIP bodies contain several others. This reviewer suggested that if the authors wish to stay with analysis of a single protein that this caveat should be made explicit under a limitations of this study section. In response, the authors have added a paragraph that details this limitation and outlines future work. This seems to be an adequate response.

Reviewer #3 also had several minor comments.

- 1) Asked about fluorescent signal saturation in the droplets and whether saturation did not lead to an underestimate of protein concentration. The authors explain that they state the signal is "60x brighter" not that "the concentration is 60x higher". They now add comment about this that I think clarify the situation.
- 2) Asked about the physiological concentration of Bik1 in cells and how the phase behavior compares to earlier reports. The authors provide an estimate of molecules per yeast cell but note that as Bik1 concentrates at microtubule ends local concentrations may be close to those used in this in vitro study. Again this seems a reasonable suggestion.
- 3) Phase diagrams for mutant Bik1s and condensate formation at lower salt concentrations. The authors now provide this information in Figure S3. Why does deleting the C-terminal QQFF motif prevent condensation? The authors add additional experiments that show this truncated construct does not form droplets even at very high protein and low salt concentrations. They add this data to Fig 1.
- 4) Clarify whether self-link peptides were found in both dispersed and condensed samples. Can the coiled coil be modified and how might this affect condensation. The authors state that proteins with these coiled coil self-links are also connected by the Cap-Gly domain. Further, it is well established that manipulating coiled coils even in a minor way can dramatically change the properties. Consequently, they argue it would be difficult to draw strong conclusions from such experiments without an extensive and dedicated study.
- 5) Comment on effective solvent properties might favor either Bik1 conformation. Were other approaches used other than cross link mass spec? The authors state they performed the suggested NMR experiments using ¹⁵N-labeled Bik1 but were unable to obtain conclusive results due to the dimeric multidomain organization of the protein.
- 6) Comment on the significance of the fractal-like network in the context of the +TIP body and add a limitations of study section. The authors already responded to this suggestion in the major comment above.

I could not play the video. It is encoded in "apcn" format and my Dell PC (Windows 10 enterprise) says it is not supported. Perhaps it could be made available in several different formats?

Version 2:

Reviewer comments:

Reviewer #2

(Remarks to the Author)

I thank the authors for their clarification, even though I am not fully sure I understand them correctly. Am I correct to assume that the authors state that all their crosslinking experiments shown in Figure 3 and 4 - including in particular the quantitative data shown in Figure 4B, 4C and 4D - were carried out as independent (biological) triplicate experiments?

The legend in Figure 4 B and C appears to be changed from “N=3 technical replicates” to “N=3”. And the critical experiment shown in Figure 4D from “Each dot represents the results for a replicate” to “Each dot represents the result of a measurement”, suggesting that those were not independent replicates but rather mere technical replicate measurements, as stated in the previous version of the manuscript.

Given the relatively large difference in measured abundances between the first and second round of experiments (Dataset 2023 vs Dataset 09.2024, Figure i, revision) the minimal differences in measured abundance between measurements/replicates shown in the current Figure 4D would be surprising, if these were indeed independent replicates. Also, even though the general trend is indeed very similar between the first and second round of experiments (Dataset 2023 vs Dataset 09.2024, Figure i, revision), some peptides – for example 311-381 - change from being depleted to being enriched.

Showing only one of two independent experiments and only parts of the data and at the same time labeling technical replicates as independent replicates - as appears to be the case in Figure 4D of the current version of the manuscript - would be grossly misleading. If this was the case, all data from both rounds of experiments (i.e. replicates) needs to be shown in Figure 4, and a third replicate is required.

In summary, if experiments were indeed carried out in independent triplicate experiments I have not further comments. Otherwise, I refer to my previous statement.

Reviewer #1

The submitted manuscript aims to dissect the structural details underpinning the phase separation of the Bik1 protein. In this reviewer's opinion, the Kar9 network is one of the more exciting systems proposed to have a phase-separation mechanism. As such, ascertaining these details is essential in understanding how these cellular processes work. The manuscript contains three themes: 1) understanding the structure of dilute phase Bik1, 2) Uncovering peptides within Bik1 that form crosslinks, and 3) Examining the mesoscale structure of Bik1 droplets. In brief, the research methods presented here are sound; the analysis generally follows the experimental results with little over-speculation, and everything is explained in an easy-to-follow narrative. In addition, the authors show the value of two underused methods (quantitative mass spectrometry and SAXS) in building a coherent, structural picture of phase separation from the single molecule to the mesoscale. Following are my comments.

I appreciate the authors' use of AF structures as the starting point and their avoidance of over-interpretation.

It needs to be made clear how the phase diagram in Fig 1D was built. I suspect that a series of samples were prepared and assessed for droplets. Perhaps I missed this, but the methods involved in this assay should be quite detailed as they are notoriously sensitive.

> We now describe in the Materials and Methods "Droplet imaging and generation of phase diagram" section of the revised manuscript how we built the phase diagram.

The data and schematic in Figure S4 should be in the main text. The data is helpful in demonstrating that homogeneity is a function of length scale, and the schematic helps convey potential structures. That said, the schematic would be a bit more useful if drawn to reflect the mass spec results.

*> As suggested by this reviewer, we have integrated both panels of **Figure S4** into a revised **Figure 1**. In our opinion, it does not make sense to show the mass spectrometry results in the schematic of **Figure 1**, as this illustration is intended to emphasize the key interaction between the CAP-Gly domain and the EEY/F-like motif of Bik1. Our Bik1 fragment studies (new **Figure S3**) indicate that this interaction plays a key role in the phase separation of the protein. However, to visualize changes in interactions between pellet and supernatant conditions, as well between phase separation and dilute, we have reported the crosslinking results in several schematics of the new **Figures 3 and 4**.*

I am typically very suspicious of DAMMIF (or similar) modeling of SAXS data. However, in this case, it seems to make sense. If it is easily accessible, I would like to see if the models fit to the raw data. My personal preference is also to see SAXS data on flexible systems presented in log-log as opposed to semi log (as the authors show in the later figures with phase separated samples). This helps accentuate the scaling in the data. While the DAMMIF models seem reasonable, I could argue that I could come to the same conclusion simply by looking at the P(r) plot combined with the rod-like scaling combined with some flexibility as a function of the length scale shown in the Kratky plot.

*> As suggested, we now show the fits of the models to the data in the revised **Figure 2A** and the SEC-SAXS data have been replotted to a log-log system.*

It is unclear how sensitive the mass spec analysis is. I'm also not sure how much sample is being measured. What I would like to know is how the thresholding for identifying a peptide relates to the likelihood of that crosslink existing. Does the crosslink need to exist in 50% of Bik1 molecules or would you detect it if it only occurred in 1% or less?

> The crosslinking experiments in this study do not provide data on the absolute fraction of peptides that are crosslinked, but they do allow for relative comparisons, showing changes in crosslinking abundance under different conditions. The thresholds used to identify confident peptides are necessary for the search engine algorithm to filter out false positive spectra and cannot be directly correlated with the number of crosslinked peptides. In principle, even crosslinks present in a small fraction of molecules, potentially below 1%, could be detected, depending on the sample's complexity and noise levels. Low-abundance peptides are less likely to have good spectra score and to be consistently detected across multiple experiments.

I appreciated the quantitative approach to the mass spec analysis presented in Figure 4. A more schematic view of how the structure evolves between dilute and phase-separated proteins would benefit the manuscript.

*> We now show schematic representations of how the structure evolves between dilute and phase separated Bik1 FL in **Figures 3 and 4**.*

It was quite interesting to me that the dilute (supernatant) sample did not clearly cluster with any of the other conditions (and seemed heterogeneous between samples). The dendrogram indicates that the supernatant is closer to the phase-separated / pellet samples than the rest. One would expect that the low ionic strength condition would cause the dimer to form some small oligomers and perhaps be missing some long-distance contacts that are sterically impossible in these smaller structures. However, these would be distinct from the high salt conformations. It would be good to see a more clear differential between these states.

*> As correctly suspected by the reviewer and based on our new XL-MS data obtained on various Bik1 truncation mutants (new **Figures 4 and S5**), we indeed think that low ionic strength conditions cause the Bik1 dimer to form some higher order oligomers prior to condensation into micrometer sized droplets. This is particularly illustrated by the new XL-MS analysis of the minimally truncated Bik1 Δ QQFF variant that revealed an intermediate behavior compared to Bik1 FL and the other, more extensively truncated Bik1 variants. At the same time, Bik1 Δ QQFF does not form any visible droplets even at very high protein (50 μ M) and very low salt concentration (50 mM NaCl; new **Figure S3**). Based on these new findings, we concluded that the interaction between Bik1's CAP-Gly domain and its C-terminal tail is key for the further condensation of higher order Bik1 oligomers into micrometer-sized droplets. We now illustrate this important finding in the main **Figure 1G** and mention it on page 5, second last paragraph, and in the Discussion section of the revised manuscript.*

I have little to nothing to say about the condensed state SAXS data other than that it is delightful to see data that looks this clean and is a model that makes logical sense and fits well. I think that the radial distribution function that came from the models is something that many SAXS experts could have drawn by eye looking at the raw data. What is interesting to me is that the author's cylinder models fit the data

well to wider angles. Given the technical nature of this manuscript, I think it would be useful to have more discussion of these features in the main text.

> Using the spherocylinder models, we have fit the data up to $q = 0.2 \text{ \AA}^{-1}$, and the spherocylinder model has enough free parameters to obtain good agreement up to this momentum transfer. This corresponds to a length of about 30 \AA , which is slightly larger than the spherocylinder diameter of 24 \AA . On this length scale, the structural details of Bik1 dimers are not captured by the spherocylinder model. The results at these length scales, therefore, are not as reliable as for larger length scales (smaller q). We now mention this in the Discussion section of our revised manuscript (page 18, second paragraph).

It would also be useful to see things like the order parameters and angle distribution not buried in the supplemental information.

*> We decided to include the order parameters and angle distributions in the supplemental information for the following reasons: (i) The order parameters correspond to model 1, while **Figure 5** presents data related to model 3. (ii) Similarly, the angle distributions pertain to models 2 and 3. However, to better streamline the supplemental content we merged **Figures S8** and **S9** into a single **Figure S8**, providing consolidated supporting information for **Figure 5**.*

In my experience, this method of measuring SAXS data for condensed phase samples shows some heterogeneity due to the volume fraction of droplets within the beam in a given integration time. Was the integration time long enough, or were the droplets sufficiently small relative to the beam size to avoid this issue? If not, it would be good to show some of the individual measurements prior to averaging. If so, it might be useful to mention in the methods.

> As suggested by the reviewer, we now show the individual measurements that were used to generate the averages shown in Figures 5A (blue curve) and in the original Figure S8A (blue, green, and red curves; now Figure S8D) in the revised version of Figure S8. We now also carefully describe our experimental setup in the Materials and Methods section of the revised manuscript (see page 23, lines 30-42).

Final minor note: the manuscript mentions the availability of the code for analyzing the mass spec data. This should include the SAXS data as well. The methods were a bit high-level with respect to how the analysis and fitting were done.

> We have deposited the code that was used to analyze the SAXS data (DOI: [10.5281/zenodo.13822678](https://doi.org/10.5281/zenodo.13822678)).

Reviewer #2

In their study Czub et al. investigate the role of the microtubule plus-end tracking protein Bik1 during phase separation. The authors express different variants of Bik1 and study their behavior in in-vitro phase separation assays by cross-linking mass spectrometry (XL-MS) and small-angle X-ray scattering (SEC-SAXS) to investigate domain organization and conformation of Bik1 and potential changes therein during phase separation. Studying the internal organization of proteins and biomolecules during phase separation is an important topic and both XL-MS and SEC-SACS are well suited techniques to experimentally address this problem.

In their manuscript the authors show an AlphaFold prediction of Bik1 together with a phase diagram of its condensation behaviour and a sketch of a number of Bik1 variants (Figure 1). However, only a subset of these variants are experimentally followed-up by SAXS (Figure 2 and 5) and only one variant is fully examined by qualitative (Figure 3) and quantitative (Fig 4) XL-MS in vitro. Here, the authors perform two sets of cross-linking experiments. In the first one, they cross-link Bik1 under high salt (dilute) and low-salt (phase separating) conditions in bulk using two complementary cross-linkers (DSS and PDH). In the second set of experiments they follow a subset of DSS induced cross-links, which they term “conformospecific” as they are only identified in either of the states, in a quantitative manner using a targeted proteomics approach.

Overall, these experiments are technically sound and carried out state of the art (of note the reviewer is no expert in SEC-SACS and therefore focuses mainly on the mass spectrometry part of the study). However, both experimental set-ups have been described previously in detail, also in the context of phase separating proteins. Moreover, the conclusions that the authors draw from the data appear in their current form overstated and require additional experiments to be supported by the data. In its current form the manuscript appears therefore too premature for publication in Nature communications and likely better suited for a more focussed journal.

> We thank this reviewer for the very valuable suggestions that significantly improved the quality and impact of our manuscript. As the reviewer will see from our extensively revised manuscript and from our answers to all reviewer's comments, we have performed a substantial number of additional experiments to solidify our initial conclusions. Specifically, we performed (i) microscopy-based phase separation experiments with Bik1 FL, Bik1 Δ N, Bik1 Δ NC, and Bik1 Δ QQFF at very high protein and low salt concentrations; (ii) reorganized and illustrated the qualitative XL-MS data on Bik1 FL (Figure 3); (iii) generated an additional quantitative XL-MS dataset quantifying conformospecific peptides in a panel of Bik1 variants at high and low sodium chloride concentration (Figure 4). Importantly, the results of these additional experiments are fully in line with all our previous conclusions made in the initial submission of our manuscript. As elaborated in more details below, they further revealed that liquid condensation of Bik1 likely involves the formation of higher-order oligomers that phase separate in a manner dependent on the interaction between the protein's CAP-Gly domain and EEY/F-like motif.

Major concerns that need to be addressed.

1. Qualitative XL-MS (Figure 3, Figure S6).

Cross-links in full-length Bik1 are compared in low salt vs high salt conditions (Fig 3). Major conclusions on domain structure, flexibility, influence of multivalency and protein networks of Bik1 in condensates are drawn from differences in single cross-links. As powerful as XL-MS is as an approach, this should not be done without additional experiments and/or follow-up studies that confirm the conclusions, as any cross-

linking information contains a certain amount of variation in itself and this variation tends to be larger within condensates and when working with proteins that contain intrinsically disordered domains.

a) The authors need to repeat these experiments using the other Bik1 constructs shown in Figure 1 - full length (FL), Δ CG, Δ tail, CC and Δ QQFF. Only by using these variants that miss domains and/or intrinsically disordered parts of Bik1, will the authors be able to confidently draw conclusions from changes in cross-linking patterns on phase separation behaviour.

*> We appreciate the reviewer's question, which prompted us to clarify a key technical aspect of the crosslinking experiment. As the reviewer correctly noted, XL-MS is subject to technical variability, and this variability is even more pronounced in heterogeneous conditions, such as during the study of conformational changes occurring during protein phase separation. Indeed, since Bik1 contains intrinsically disordered regions (L1 and L2; **Figure 1**) and forms a dimer, the protein can explore multiple conformational states. At low salt concentrations, Bik1 may constantly transition between the dilute and condensed phases, and since the crosslinking reaction is not selective for a single state, the XL-MS approach captures a heterogeneous system (DOI: 10.1038/s41596-023-00900-0).*

To address this limitation, we thought to apply a reductionist approach consisting of two stages: (i) First, we identified crosslinked peptides unique to one condition (conformospecific crosslinks). (ii) Second, we repeated the experiment using targeted proteomics to quantify conformospecific peptides in a panel of Bik1 variants. This reductionist strategy simplifies the description of conformational changes during phase separation, although it may not capture all conformational states and interactions.

*In Step 1 (qualitative XL-MS, **Figure 3**), we analyzed the XL-MS data to generate a library of peptides representing specific conformational states. The identification of condition-specific crosslinks is illustrated in the upset plot shown in **Figure 3B**, where we identified 14 crosslinked peptides unique to the pellet condition, 9 specific to pellet and phase-separated conditions, and 4 specific to the supernatant and dilute conditions. We then refined this peptide list by excluding peptides that are difficult to identify via targeted proteomics, particularly longer crosslinked peptides or those with unambiguous spectral attributions. In Step 2 (quantitative XL-MS, **Figure 4**), we quantified 17 conformospecific peptides across a panel of Bik1 variants. The strength of this approach lies in its ability to identify conformational states by analyzing the full-length protein, thereby generating a library of Bik1 conformational states that can be exploited to characterize truncation mutants.*

*In the revised manuscript, we modified the section on generating the crosslinked peptide library and expanded it to include self-links (207-207, 311-311, and 374-374), as these crosslinks provide insights into the oligomeric state of Bik1 (**Figure 3**). For the generation of the library all conditions have been acquired in triplicate with the exception of the pellet condition. To further demonstrate the applicability of this approach, we expanded the dataset and conducted three additional experiments: (i) We showed that all crosslinked peptides identified in the pellet condition in **Figure 3** are also enriched in the targeted analysis. In the new version of the manuscript, we performed the experiment in triplicate using a different batch of protein and instrumentation, demonstrating the robustness of the approach (**Figure 4B**). (ii) We demonstrated that crosslinked peptides identified in the pellet condition are also enriched in the phase-separated condition compared to the dilute condition, indicating that the mechanism driving phase separation is conserved. Notably, despite using a different batch of protein purification and a different MS setup (see above), we were able to replicate the results from the initial version of the manuscript, consistently showing enrichment of the peptides 23-374, 79-338, 211-359, 216-245, and 211-346 (see*

Figure i for the reviewers). (iii) We extended this analysis by examining the abundance of conformer-specific peptides in a panel of Bik1 truncation variants at high and low salt concentrations, showing a strong correlation between droplet assays and the crosslinked peptide analysis (Figure 4D).

Figure i: Volcano plot comparison illustrating the differential abundance of conformer-specific peptides between the phase-separated and dilute conditions in the previous experiment (left) and the new experiment batch (right).

b) Their approach contains a mix of Bik1 in condensates and in the dilute phase (as a significant portion of Bik1 under low salt conditions will not partition into condensates). The authors therefore need to compare their data to the pellet and supernatant fraction in Figure S6, which should also consist of Bik1 in the condensed and dilute phase. However, even from a quick glance at the data it is obvious that there are large discrepancies between the datasets that should contain the same sample type – i.e. 3A (dilute) and S6 (supernatant) or the condensed forms 3A (phase separated) and S6 (pellet). The authors need to explain this discrepancy.

> *It is possible to compare the consistency of results between the phase-separated and dilute conditions, as well as between the pellet and supernatant conditions, both qualitatively (Figure 3A) and quantitatively (Figure 4C). In the revised manuscript, we emphasized the qualitative comparison in Figure 3A, where we show that residues in the phase-separated and pellet conditions are much more interconnected and characterized by long-range interactions compared to the crosslinked peptides in the dilute and supernatant conditions. The quantitative analysis provides even stronger evidence of the similarity between the pellet and phase-separated conditions, demonstrating that the interactions enriched in the pellet versus supernatant are also conserved in the phase separation versus dilute conditions (207-207, 211-359, 211-346, 216-245, 79-338, 23-374; Figure 4C). Finally, the unsupervised hierarchical clustering of the quantitative data (Figure ii for reviewers and Figure S6C) further supports this notion, grouping the supernatant and dilute conditions (500 mM NaCl) together and the pellet and phase-separated conditions (150 mM NaCl) together.*

Figure ii. Unsupervised hierarchical cluster based on Pearson correlation for the conformospecific peptides for the following Bik1 conditions: phase separation (Bik1_FL_150), pellet (Bik1_FL_Pellet), supernatant (Bik1_FL_SN) and dilute (Bik1_FL_500) conditions.

c) Δ QQFF Bik1 is probably the most interesting variant of the study – as four missing amino acids preclude phase separation. The authors should make further sub-constructs and also test different salt conditions under which Δ QQFF phase separates.

> Based on a comment of reviewer 3, we have performed phase separation experiments with all our Bik1 fragments, i.e., Bik1 Δ N, Bik1 Δ C, Bik1 Δ NC, and Bik1 Δ QQFF. Remarkably, as shown in the new **Figure S3** of the revised manuscript, none of the four Bik1 fragments did form droplets even at a ten-fold higher protein concentration (50 μ M) and five-fold lower sodium chloride concentration (50 mM) as needed for Bik1 FL condensation. Based on this finding, we felt that the testing of additional sub-constructs would not add much to the current story.

d) In order to draw robust conclusions on protein-protein contacts – as for example insinuated by the title of the manuscript – the authors should not only use “selflinks” but need to repeat these experiments by mixing labelled and unlabelled forms of Bik1

> The idea of the reviewer of mixing, e.g., isotopically labeled and unlabeled forms of Bik1 is indeed an excellent one since it would enable us to distinguish intra- from inter-molecular contacts of the dimeric protein in a straightforward manner. This approach has been used in the context of phase separation to show that alpha-syn monomers are in elongated phase separation transitioning to phase separation condition (DOI: 10.1002/anie.202205726). However, in case of Bik1, it is known for a long time that two-stranded coiled-coil proteins can readily self-exchange their chains (DOI: 10.1002/bip.360310805). Equimolar mixing of labeled and unlabeled Bik1 molecules could thus result in a heterogeneous distribution

of homo and hetero labeled dimeric species. To perform experiments allowing us to draw robust conclusions we thus would need to experimentally assess the chain exchange kinetics under the several solution conditions that were used to perform our XL-MS experiments, which we feel goes beyond the scope of the present study.

2. Quantitative XL-MS data (Figure 4).

a) Also in this case the experiments need to be repeated using additional Bik1 variants, see above. Critically, conditions under which Δ QQFF starts to phase separate should be investigated and included.

*> As suggested by the reviewer, we have repeated the quantitative XL-MS experiments with Bik1 FL and its truncation mutants. As shown in the new **Figures 4 and S6**, we reproduced the initial results obtained with Bik1 FL. We further found that while extensively truncated Bik1 versions did not show any significant differences between dilute and phase separated samples, the minimally truncated version Bik1 Δ QQFF behaved somewhat in between, indicating the formation of higher order oligomers. Since our new phase separation experiments demonstrate that Bik1 Δ QQFF does not form any visible droplets even at very high protein (50 μ M) and very low salt concentration (50 mM NaCl; new **Figure S3**), we concluded that the interaction between Bik1's CAP-Gly domain and its C-terminal tail is key for the further condensation of higher order Bik1 oligomers into micrometer-sized droplets. We now illustrate this important finding in **Figure 1G** and mention it on page 5, second last paragraph, and in the Discussion section of the revised manuscript.*

b) The comparison of the existing full-length data itself appears, at least partly, inconclusive. Why do cross-links within/between the same domains show opposite behaviour (Figure 4D)? For example – why are the links 7-359 and 16-338 down regulated but 23-374 or 79-338 upregulated? (This must not necessarily be a problem. However, if the authors, as they do in the current form of the manuscript, draw conclusions from changes in single cross-links during phase separation, then the authors need to explain).

> The differing abundance of crosslinked peptides within or between the same domain reflects the highly dynamic and heterogeneous nature of proteins as they transition from dilute to phase separation. Specifically, Bik1 is characterized by significant flexibility, with over half of the protein (regions 1-189 and 386-440) exhibiting a high pLDDT score (>90). This flexibility allows the Bik1 dimer to explore multiple conformational states, and during the phase separation transition, it may adopt distinct conformations or interactions under varying conditions. While these results may appear contradictory, they likely indicate that certain regions become more accessible or favorably positioned for crosslinking.

In addition to this general consideration, we would like to point out that despite repeating the experiment with a different batch of protein, different batches of XL reagents, and using a different MS setup, we consistently observed upregulation for peptides 23-374 and 79-338, with no significant change for peptide 7-359.

c) As above, the authors need to explain why differences between samples that should contain the same state (i.e. dilute/supernatant vs pellet/phase sep are at least partly in larger disagreement than samples that should contain different states (Figure 4C).

> We addressed this point in our response to point 2.1b of this reviewer. In the revised version of our manuscript, we have included a qualitative comparison of crosslinked peptides in **Figure 3A**, along with two side-by-side volcano plots comparing pellet versus supernatant and phase separation versus dilute conditions (**Figure 4**, panels **B** and **C**, respectively).

d) Figure 4 panels 4C and D are quite confusing. Similar links should be mapped onto the structure or the 2D model – as in Figure 3 – and up- and down regulated links should be indicated.

> For **Figure 4** we have generated a new dataset and, based on the comment of the reviewers, we presented the results differently. In the revised manuscript, we display the significant changes in the protein's primary sequence below the volcano plot.

3. Comparison between SAXS and XL-MS data. There is very little comparison between the datasets.

> SAXS is a method of relatively low resolution that can examine the nanoscale conformation of biomolecules. By combining SAXS data with experimental or theoretical structures, this technique becomes a powerful tool for determining the averaged shapes of proteins in dilute solutions as well as their arrangement in condensed phases. XL-MS on the other hand offers a higher resolution approach, defining protein-protein interactions at the single residue level and probing the conformational states of proteins in different solution conditions. When combined, SAXS and XL-MS can provide a comprehensive view of the structure, dynamics, and interactions of proteins, offering both global and local insights. We now mention this comparison on page 18, lines 38-48 of the revised manuscript.

Minor Comments

Was the filtering strategy only based on spectra quality (Id.Score 25 and FDR < 0.05), as written in the methods, or also on consistency (crosslinked peptides must be identified in at least two out of three replicates) – see page 8 of the manuscript? This is quite an important information and should be clarified.

> In the revised manuscript, we have clarified the filtering strategy adopted for the analysis of crosslinked peptides in both the Results and Materials and Methods sections: "DSS Crosslinked peptides with Id.Score > 20 and filtered for consistency (identification in at least two out of three replicates) were considered in the analysis. PDH Crosslinked peptides with Id.Score > 25, deltaS > 0.9 and FDR < 0.05 were considered in the analysis."

The colors in Figure 4D are very hard to see.

> **Figure 4** has been regenerated based on the results of the newly acquired XL-MS datasets.

The tile appears to be an overstatement and should be changed.

> The title "Phase separation of a microtubule plus-end tracking protein into a fluid fractal network" is primarily motivated by the results obtained from small-angle X-ray scattering (SAXS) of the condensed

phase (Figure 5, Movie 1). It does not imply any specific details regarding the interactions occurring within the fractal network.

Reviewer #3

This study investigates the phase separation and internal organization of Bik1, a microtubule plus-end tracking protein (+TIP) from the CLIP-170 family, which plays a crucial role in budding yeast cell division. The authors demonstrate that Bik1 exists as a rod-shaped dimer, with its conformation predominantly influenced by a central coiled-coil domain. During the process of liquid condensation, Bik1 undergoes significant conformational rearrangements, resulting in a 2-3-fold increase in interactions between the protein's folded and disordered domains. This highlights a complex interplay between different structural regions of the protein. This paper also suggests that the supramolecular structure of Bik1 condensates is not homogeneous like classical liquids, but rather displays a heterogeneous, fractal architecture characterized by distinct protein-rich and protein-free domains. The detailed insights from this study shed light on the physical properties and dynamic rearrangement of the Bik1 protein in both its dilute and condensed phases.

Major Comment

In recent years, +TIPs have garnered considerable attention due to their role in several microtubule-based cellular processes, with a recent proposition that they function as liquid condensates forming dynamic, membrane-less organelles. However, the detailed mechanisms governing the formation and internal organization of these condensates are yet to be answered as previous studies were primarily focused on the functional aspects of +TIP liquid-like condensates in microtubule dynamics. The experimental approach used here can be applied to other +TIP networks, potentially revealing new aspects of the organizational principles underlying microtubule-associated proteins.

While these techniques hold promise for robust structural characterization of proteins within condensates, this work is limited by its description of just a single component of the +TIP body, whereas it is already known that there are several other protein components such as Kar9, Bim1, among others, undergoing multivalent network interactions resulting in the formation of the liquid-like +TIP condensate (as also mentioned by the authors). Currently, the field is moving towards more physiological reconstitutions, and reconstituting a more physiologically relevant +TIP body and then assigning these interactions and studying the change in the conformational landscape of each of proteins would be more relevant and informative towards understanding its role in modulating microtubule dynamics. If they want to stick to one protein, it's important to add this caveat under a limitations of the study section, and not overclaim on the relevance.

> Indeed, while our study offers the techniques for robust structural characterization of proteins within condensates, we currently can only speculate about the implications of our in vitro findings obtained on a single protein for +TIP body behavior and function in budding yeast cells. One distinctive property of Bik1 is that it can phase separate into liquid droplets on its own, which contrasts with the other two core +TIP body components Kar9 and Bim1, which separately form aggregates or do not phase separate, respectively (DOI: 10.1038/s41556-022-01035-2). Bik1, and its C-terminal EEY/F-like motif QQFF in particular, seems thus a key component determining the materials properties of the +TIP body as a whole. Whether the fractal structure of Bik1 condensates is also present in Kar9-Bim1-Bik1 droplets needs to be assessed experimentally in follow up studies. However, if this were indeed to be the case then the achieved high viscosity at relatively low protein concentration of +TIP body droplets could indeed offer the glue-like, mechanical-coupling properties to drive nuclear positioning during budding yeast mitosis and mating. Based on the reviewer's recommendation, we have added such a "limitations of the study" paragraph to the Discussion section of our revised manuscript (page 18, lines 28-37).

Minor Comments

1. “These fluorescence microscopy experiments suggest that the distribution of Bik1 dimers is uniform throughout the droplet; the fluorescence of Bik1 is over 60 times brighter in the condensed than in the dilute phase (Figures 1E and S4A). “

Did the authors perform control experiments to ascertain that signal saturation did not cause underestimation of protein concentrations using fluorescence-based measurements.

*> We did not do those control experiments. That is why we were careful to say “the fluorescence is 60x brighter” instead of saying “the concentration is 60x higher”. None of our conclusions use this value. Nevertheless, we thought it would be interesting to explore the referee’s hypothesis. We independently estimated the concentration of Bik1 using two different label-free methods. First, the peak of $g(r)$ in the dense phase is at a distance of about 20 nm. This corresponds to a concentration of about 1 mM. Second, the density mismatch between the dense and dilute phase reported in one of our previous papers (DOI: 10.1039/d0sm01319f) was found to be about 30 kg/m³. Using the law of mixtures for density, a typical protein density of 1350 kg/m³, and Bik1 volume of about 135 nm³ (from the sphero-cylinder model), this also gives a dense phase concentration of 1 mM. In the phase diagram in **Figure 1D**, we measure a dilute phase concentration of Bik1 of 2.5 μ M. Together, these measurements suggest that the concentration ratio in the dense and dilute phases is about 400, a value that is much larger than suggested by the fluorescence measurement. Therefore, the referee is correct that there is either a large amount of fluorescent saturation in the dense phase or the fluorophore interferes with the phase behavior. We added a statement about the dense phase concentration on page 4, line 43 until page 5 line 2 of our revised manuscript.*

2. The authors provide a phase diagram for the condensation of Bik-1 protein as a function of salt concentration where the protein seems to exhibit a threshold concentration of around 5 μ M at 250 mM NaCl (the concentration of salt chosen for experiments in this work).

Can the authors comment on what is the physiological concentration of this protein and how it relates to this condensation observed at μ M range of protein concentrations. Also, how does the phase behaviour compare to what was reported earlier for Bik1 in the Multivalency ensures persistence of a +TIP body at specialized microtubule ends | Nature Cell Biology (Maier et al. Nat. Cell Biol. 2023). (Ref 14 in the manuscript).

> The number of Bik1 molecules was estimated to be around 2000 copies per budding yeast cell (DOI: 10.1016/j.cels.2017.12.004). Assuming an average volume of a budding yeast cell of 50 μ m² (DOI: 10.1073/pnas.0901851107) and considering the space that organelles occupy, we can estimate a concentration for Bik1 dimers of around 100 nM, which is 1-2 orders of magnitude lower than the micromolar concentration range used in our different in vitro studies. However, Bik1 gets concentrated at dynamic microtubule plus ends, an effect that could indeed lead to a concentration range that is close to the one used in our studies. We now mention this on page x, last paragraph.

In our previous Maier et al. Nature Cell Biology 2023 paper (DOI: 10.1038/s41556-022-01035-2), we did not generate a phase diagram for Bik1. We thus cannot compare the new data in the present manuscript with our previous work. However, we used the same construct as well as the protein expression, purification, processing, and phase separation protocols as described in our former study. We now mention this in the Materials and Methods “Protein purification” section of the revised manuscript.

3. “Deletion of the CAP-Gly or the zinc-finger-containing domain (Bik1 Δ CG and Bik1 Δ tail, respectively) both suppressed protein condensation at low salt conditions (e.g., 250 mM sodium chloride) where Bik1 FL readily formed droplets. Therefore, we investigated whether mixing the two truncated variants under low salt conditions can rescue their potential to undergo condensation. 5 However, no visible droplets were observed under the same conditions used for Bik1 FL (Figure S3).....This finding indicates that the introduced truncations in both Bik1 variants reduced the multivalency of the system needed for efficient phase separation of Bik1 FL. Additionally, we examined the relevance of the C-terminal EEY/F-like motif QQFF of Bik1 for the protein’s ability to undergo condensation. Interestingly, the deletion of the motif in Bik1 FL (Bik1 Δ QQFF; Figure 1C) was sufficient to prevent phase separation of this minimally truncated Bik1 variant (Figure S3)”

Have the authors tried to construct phase diagrams, like the WT Bik1, for the rest of the mutants? Is there a possibility that these mutants could still form condensates at lower salt concentrations (< than 250 mM NaCl) as the FL has several regions of positive and negative charges. Also, the authors have used a 250 mM NaCl solution for comparison of condensate formation propensity but have resorted to using a lower salt concentration in their cross-linking mass spectrometry experiments. Is there any reason for that?

*> We have performed phase separation experiments with all our Bik1 fragments, i.e., Bik1 Δ N, Bik1 Δ NC, and Bik1 Δ QQFF. Remarkably, as shown in the new **Figure S3** of the revised manuscript, none of the four Bik1 fragments did form droplets even at a ten-fold higher protein concentration (50 μ M) and five-fold lower sodium chloride concentration (50 mM) as needed for Bik1 FL condensation. Based on these observations, we decided to use an intermediate salt concentration for the XL-MS experiments.*

Also, as the authors already highlight how the deletion of just 4 C-terminal residues (construct Δ QQFF) prevents condensation completely at the experimental conditions used in this work, can the authors comment on what might be the cause. Interestingly, the authors mention on Page 11 Lines 361-364, that the relative abundance of specific crosslinks for the Bik1 Δ QQFF were found to be somewhere in between the pelleted phase-separated fraction and the dilute dispersed fraction of WT Bik1, thereby, suggesting that formation of certain higher order oligomers in principle prevented the condensation of this construct. It would be interesting if the authors could characterize these oligomeric intermediates (if formed) and then how they prevent protein condensation.

*> Based on our previous work establishing CAP-Gly domains as EEY/F-motif binding domains (DOI: 10.1016/j.molcel.2006.07.013, DOI: 10.1038/nsmb1291, DOI: 10.1016/j.str.2018.03.003, DOI: 10.1016/j.jsb.2011.11.010, DOI: 10.1016/j.tibs.2008.08.006), we concluded that deleting the EEY/F-like motif in Bik1 (Δ QQFF construct) abrogates the interaction between the protein’s CAP-Gly domain and its C-terminal tail (**Figure 1G**). As correctly recognized by the reviewer, our new XL-MS data (**Figure 4, Figure S6**) indicate that Bik1 Δ QQFF can indeed form certain higher order oligomers. However, our new phase separation experiments demonstrate that Bik1 Δ QQFF does not form any visible droplets even at very high protein (50 μ M) and very low salt concentration (50 mM NaCl; **Figure S3**). Based on these new findings, we concluded that the interaction between Bik1’s CAP-Gly domain and its C-terminal tail is key for the further condensation of higher order Bik1 oligomers into micrometer-sized droplets. We now illustrate this important finding in **Figure 1G** and mention it on page 5, second last paragraph, and in the Discussion section of the revised manuscript.*

4. “Four out of five selflink peptides were identified in the coiled-coil domain (Lys207, Lys245, Lys263, and Lys374)”

Were these interactions between the CC domain residues found in both the dispersed and condensed phases or are there any changes in this distribution to accommodate interactions with the CAP-Gly domain in the condensed phase? Also, can the authors speculate or show how the condensation behaviour might be affected by deleting or modifying sections of the CC domain either within the heptad repeats or the disordered linker sequences?

> The XL-MS experiments show that during the transition to phase separation, Bik1 undergoes structural rearrangement, with enhanced interactions between the coiled-coil region and between the coiled-coil region and the CAP-Gly domain. Since in the phase separation condition, the coiled-coil residues that harbor self-links (207, 263, and 374) are also connected with the CAP-Gly domain (Figure 3A), we hypothesize that these interactions may be mutually exclusive. It is likely that, in both phase separation and pellet conditions, there are distinct populations of Bik1 conformers, including at least one where the coiled-coil region interacts with the N-terminal region and another where the coiled coil interacts with itself.

In our opinion, it is very difficult to predict how deleting or modifying sections of the coiled-coil domain, within or between heptad repeats, would affect Bik1 condensation. In this context, it should be noted that it is very well established that deletions or even small modifications of coiled-coil domains can result in an abrogation of coiled-coil formation, a change in oligomerization state, and/or a change in the orientation of the coiled-coil chains (DOI: 10.1093/emboj/17.7.1883; DOI: 10.1073/pnas.1008502107; DOI: 10.1021/bi981269m; DOI: 10.1016/j.jbc.2023.104579). Thus, any mutations introduced in Bik1’s coiled coil would need to be carefully analyzed in terms of structural integrity, stability, conformational properties, and interaction properties to derive meaningful conclusions. We feel that such a completely new line of experimentation goes beyond the scope of the present manuscript but would indeed deserve a dedicated study aiming at understanding the role of coiled-coil domains in protein phase separation.

5. “Monolinks with reduced abundance levels in the pellet sample (Lys310 and Lys373). They can serve as proxies for assessing residue accessibility. This result suggests that in the Bik1 pellet condition, the coiled-coil region around Lys310 and Lys373 is much less exposed to the solvent than in other conditions. Differences in the coiled-coil residue exposure to the solvent or other Bik1 domains might be related to the presence of the kinks between coiled-coil segments,”

Can the authors comment on how the effective solvent property within the condensed phase might favour either of the Bik1 conformations: “rod-like or V-shaped”. Apart from using this technique of cross-linking mass spectrometry, did the authors ever consider investigating the shift in the conformational paradigm of Bik1 in transitioning from the dispersed to the condensed phase, using site-specific fluorescence, NMR, and/or EPR-based measurements?

> We indeed performed NMR experiments with ¹⁵N-labeled Bik1; however, no conclusive results could be obtained due to the size and complexity of the dimeric, multidomain nature of the protein. We did not consider performing site-specific fluorescence- or EPR-based experiments so far but agree that these are excellent ideas to tackle the conformational paradigm of Bik1. Along similar lines, one could also envisage using computational approaches (e.g., stirred MD simulations) to get insight into this issue in a comprehensive biophysical follow up study.

6. It is interesting to note the formation of a fractal-like network within the Bik1 condensates but can the authors comment on what is the exact significance of this sort of a network in the context of +TIP body and what could be its role in modulating microtubule dynamics. In particular, given that this is a single protein, the authors should have a paragraph on limitations of the study. The authors can expand on this and highlight its relevance in the biological context.

> We have included a "limitations of the study" paragraph in the revised version of our manuscript (see page 18, lines 28-37).

Reviewer #1

The revised manuscript has addressed all of my suggestions. I recommend it for publication.

> *We thank the reviewer that they were satisfied with our revisions.*

Reviewer #2

The authors have significantly improved the quality of their manuscript by adding new data and addressing important issues raised by the reviewers. In particular, by adding an additional quantitative XL-MS dataset for conform-specific peptides in a panel of Bik1 variants (Bik1 FL, Bik1 DN, Bik1 DNC, and Bik1 DQQFF) at high and low sodium chloride concentration the authors have clearly strengthened their manuscript.

Unfortunately, fundamental points, which are central for their line of argumentation have not been addressed and thus the manuscript still requires additional revision before it can be accepted for publication.

Specifically, the cross-linking data is based on single (biological) measurements, as only technical replicates have been conducted. This is true for both the qualitative as the quantitative analysis (see Figure 3 and 4, in particularly, Figure 3A and Figure 4B, C and D).

While for the qualitative analysis one may argue that by employing different linkers a somewhat comprehensive picture has emerged, this remains a central flaw of the quantitative data.

As argued before, the authors need to include true (biological) replicates, both to identify their list of conform-specific crosslinks (i.e. Figure 4A and 4B) AND then to conduct their quantitative analysis for their subset of conform-specific crosslinks for the panel of Bik1 variants (Figure 4D).

Data from mere technical replicates is not sufficient to allow for any robust interpretation of changes in crosslink abundances and thus protein dynamics, in particularly for such dynamic systems as phase separated proteins.

> We apologize for not having adequately explained and correctly quoted the preparation of our sample replicates. To clarify, the XL-MS experiments were performed with three independent experimental replicates and not with technical replicates as wrongly mentioned in several instances of the manuscript. Specifically, we induced condensate formation in separate tubes, and for each tube, we conducted separate crosslinking reactions followed by independent bottom-up MS sample preparations. Therefore, these are indeed independent experimental replicates.

Notably, for the quantitative experiments originally requested by Reviewer #2, we repeated the analysis on Bik1 FL alongside all additional Bik1 variants (Figure 4). For this experiment, we generated and analyzed a new XL-MS dataset that was acquired with the following variations:

- 1. The experiments were conducted one year after the initial dataset was collected.*
- 2. Different batches of purified proteins were used.*
- 3. Different buffers were used to induce protein phase separation.*
- 4. A different researcher (T.G. instead of M.P.C.) performed the protein purification.*
- 5. A new batch of crosslinking reagents was employed.*
- 6. A completely different MS setup was used, including a different MS instrument and an alternative chromatographic system, due to the relocation of F.U. from ETH Zürich to the University of Mainz.*

Despite these substantial changes in experimental conditions, the consistency of our results underscores the robustness of our approach. By analyzing conformationally specific peptides and supporting the data with rigorous targeted quantification and statistical analysis, we confirmed the findings from the initial version of the manuscript. Specifically, we observed consistent enrichment of peptides 23-374, 79-338, 211-359, 216-245, and 211-346 (Figure i for the reviewer), demonstrating the reliability and reproducibility of our methodology.

We apologize again for any confusion this oversight may have caused and thank the reviewer for identifying this important issue. We have now clarified throughout the revised manuscript that the XL-MS experiments were performed with independent experimental replicates.

Figure i: Volcano plot comparison illustrating the differential abundance of conformospecific peptides between the phase-separated and dilute conditions obtained in the initial (left) and the new experiment (right).

As a side note. It is very hard to follow the revisions of the authors as no version of the revised manuscript with tracked-changes has been submitted.

> We now submitted a revised version of our manuscript in which the major changes of the first round of revisions are highlighted in green and the ones from the second round in cyan.

Reviewer #3

Reviewer #3 had one major comment concerning the focus on a single protein (Bik1) when +TIP bodies contain several others. This reviewer suggested that if the authors wish to stay with analysis of a single protein that this caveat should be made explicit under a limitations of this study section. In response, the authors have added a paragraph that details this limitation and outlines future work. This seems to be an adequate response.

Reviewer #3 also had several minor comments.

1) Asked about fluorescent signal saturation in the droplets and whether saturation did not lead to an underestimate of protein concentration. The authors explain that they state the signal is “60x brighter” not that “the concentration is 60x higher”. They now add comment about this that I think clarify the situation.

2) Asked about the physiological concentration of Bik1 in cells and how the phase behavior compares to earlier reports. The authors provide an estimate of molecules per yeast cell but note that as Bik1 concentrates at microtubule ends local concentrations may be close to those used in this in vitro study. Again this seems a reasonable suggestion.

3) Phase diagrams for mutant Bik1s and condensate formation at lower salt concentrations. The authors now provide this information in Figure S3. Why does deleting the C-terminal QQFF motif prevent condensation? The authors add additional experiments that show this truncated construct does not form droplets even at very high protein and low salt concentrations. They add this data to Fig 1.

4) Clarify whether self-link peptides were found in both dispersed and condensed samples. Can the coiled coil be modified and how might this affect condensation. The authors state that proteins with these coiled coil self-links are also connected by the Cap-Gly domain. Further, it is well established that manipulating coiled coils even in a minor way can dramatically change the properties. Consequently, they argue it would be difficult to draw strong conclusions from such experiments without an extensive and dedicated study.

5) Comment on effective solvent properties might favor either Bik1 conformation. Were other approaches used other than cross link mass spec? The authors state they performed the suggested NMR experiments using ¹⁵N-labeled Bik1 but were unable to obtain conclusive results due to the dimeric multidomain organization of the protein.

6) Comment on the significance of the fractal-like network in the context of the +TIP body and add a limitations of study section. The authors already responded to this suggestion in the major comment above.

> We thank the reviewer that they were satisfied with our revisions.

I could not play the video. It is encoded in “apcn” format and my Dell PC (Windows 10 enterprise) says it is not supported. Perhaps it could be made available in several different formats?

> We now submitted the movie in MP4 format, which should run on a Windows 10 machine.

Reviewer #2

I thank the authors for their clarification, even though I am not fully sure I understand them correctly. Am I correct to assume that the authors state that all their crosslinking experiments shown in Figure 3 and 4 - including in particular the quantitative data shown in Figure 4B, 4B and 4D - were carried out as independent (biological) triplicate experiments?

> We apologize for not making this point sufficiently clear earlier. We fully confirm that all the experiments presented in Figure 4 were conducted in three independent (biological) replicates. Additionally, we affirm that all data depicted with error bars were in general derived from three independent experiments.

The legend in Figure 4 B and C appears to be changed from “N=3 technical replicates” to “N=3”. And the critical experiment shown in Figure 4D from “Each dot represents the results for a replicate” to “Each dot represents the result of a measurement”, suggesting that those were not independent replicates but rather mere technical replicate measurements, as stated in the previous version of the manuscript.

> We confirm that each dot shown in Figure 4D stems from an independent experiment. We have rephrased the sentence accordingly.

Given the relatively large difference in measured abundances between the first and second round of experiments (Dataset 2023 vs Dataset 09.2024, Figure i, revision) the minimal differences in measured abundance between measurements/replicates shown in the current Figure 4D would be surprising, if these were indeed independent replicates.

> Again, we fully confirm that we used independent experimental replicates for assembling both datasets, Dataset 2023 and Dataset 2024. Notably, the variability observed in the independent experimental replicates shown in Figure 4D (Dataset 2024) is comparable to that in the previous dataset (Dataset 2023): The median coefficient of variation (CV) for each crosslinked peptide and condition is 0.242 and 0.259 for the Dataset 2023 and Dataset 2024, respectively. This contrasts with the expected CV value for technical replicates that typically fall significantly lower within the range of 0.05 to 0.1 (PMID: 35767427).

Also, even though the general trend is indeed very similar between the first and second round of experiments (Dataset 2023 vs Dataset 09.2024, Figure i, revision), some peptides – for example 311-381 - change from being depleted to being enriched.

> In the Volcano plot, the crosslinked peptide 311-381 does not show a significant change (p -value > 0.05 in both datasets). A change of 0.2 on the \log_2 scale corresponds to a difference of approximately 10%, which is below the variability of the dataset (around 25%, see above). Therefore, the abundance of this peptide did not change significantly, and we consequently did not use it to describe the protein rearrangement between dilute and phase separation conditions.

Showing only one of two independent experiments and only parts of the data and at the same time labeling technical replicates as independent replicates - as appears to be the case in Figure 4D of the current version of the manuscript - would be grossly misleading. If this was the case, all data from both rounds of experiments (i.e. replicates) needs to be shown in Figure 4, and a third replicate is required.

> As mentioned above, we fully confirm that the data presented in Figure 4D are derived from independent replicates. Since we did not include data on specific Bik1 mutants during the initial round of experiments (i.e., Dataset 2023) and given that we conducted three independent experimental replicates for each condition across Dataset 2024, we believe it is sufficient to show only latter dataset.

In summary, if experiments were indeed carried out in independent triplicate experiments I have not further comments. Otherwise, I refer to my previous statement.

> Based on how we performed our experiments, we can fully reassure the reviewer that all data depicted with error bars were derived from three independent experiments.